# Beyond Shortest-Paths: A Benchmark for Reinforcement Learning on Traffic Engineering

## Abstract

Selecting efficient routes for data packets is an essential task in computer networking. Given the dynamic of today's network traffic, the optimal route varies greatly with the current network state. Despite the wealth of existing techniques, Traffic Engineering in networks with changing conditions is still a largely unsolved problem. Recent work aims at replacing Traffic Engineering heuristics with Reinforcement Learning but does not provide a formalism that covers all challenges of Traffic Engineering, or a reference framework for training and evaluating under realistic network conditions in a reproducible manner. We fill these two gaps by casting distributed Traffic Engineering as a Swarm Markov Decision Process, and introducing a training and evaluation framework powered by a faithful network simulation engine that implements it. Using our framework, we further train and evaluate two policies on a large variety of scenarios to showcase the effectiveness and versatility of our framework. Our experiment results expose the weaknesses of existing routing protocols and highlight the difficulty of this open problem.

## 1 Introduction

In computer networks, Routing Protocols (RPs) find paths between nodes. These paths can be optimized for, e.g., high throughput, low latency, low packet loss rate, or low resource utilization, and thus Routing Optimization (RO) plays a key role in Traffic Engineering (TE) (Wang et al., 2008). The heuristics of existing RPs such as Open Shortest-Path First (OSPF) (Moy, 1997) or Enhanced Interior Gateway Routing Protocol (EIGRP) (Savage et al., 2016) work well in static scenarios, but computer networks often are unpredictable and dynamic. Erratic traffic demands, failing hardware and constantly changing user requirements may require route re-optimization within a few milliseconds to prevent sharp drops in performance (Gay et al., 2017a). Particularly for previously unseen network situations, heuristics have to be manually adjusted, which takes time and often manual labor. One naive solution is to sweepingly increase network capacity in regions where congestion is regularly observed, but network over-provisioning is costly and highly inefficient because network components by design stay far below their capability limits for the majority of the time. Instead, an ideal RP provides optimal routing paths at any point in time, and regardless of the network topology *as well as* current utilization and traffic situation (Avin & Schmid, 2019). For the RO mechanism of such RPs (which we call *general-purpose RO*), we identify the following requirements:

1. **Timeliness:** Network performance issues like congestion cause packet delays and drops if no counter-measure is taken. As the loss of data and service quality increases over time, routing decisions become weaker the longer it takes to calculate and install them on the network. This qualifies RO for TE as a soft real-time system (Marchand et al., 2004).

2. **Compatibility:** Protocol extensions like ECMP or MPLS (Iselt et al., 2004) or overlay techniques like Segment Routing (SR) (Filsfils et al., 2018) can complement RPs to meet performance requirements, but add extra complexity and interference effects to the overall network setup. Drop-in replacements for RPs can serve as an alternative that is easier to deploy (Bernárdez et al., 2023).

3. **Generality:** Real networks vary greatly in topology and configuration (e.g. link data rates, processing delays, buffer sizes). The learned model should be able to optimize the routing of any network, no matter its topology or configuration.

4. **Robustness and Resilience:** In the face of expected and unexpected events such as a planned change of network topology/configuration or local/regional network failures, the learned model should be able to adapt the routing if necessary.

5. **Scalability:** With increasing network scale, centralized RO approaches become less and less useful due to longer communication pathways. While it can be beneficial to logically divide networks into smaller units for some tasks, locally optimal routing is not guaranteed to be globally optimal (Dietterich, 2000). Decentralized and distributed RO approaches can be designed to deal with large networks, but require efficient communication strategies.

6. **Realism:** Evaluation settings that rely on analytical paradigms like network calculus (Le Boudec & Thiran, 2001) or queueing theory (Newell, 2013) work with a greatly simplified network model that rules out the complex system dynamics caused by protocol and component interplay. On the other hand, the required variety and amount of training data make training complex models entirely on real networks prohibitively expensive. Discrete-event simulators like ns-3 (Henderson et al., 2008) can provide a middle-ground, combining affordable and repeatable evaluation with versatile and faithful network modeling.

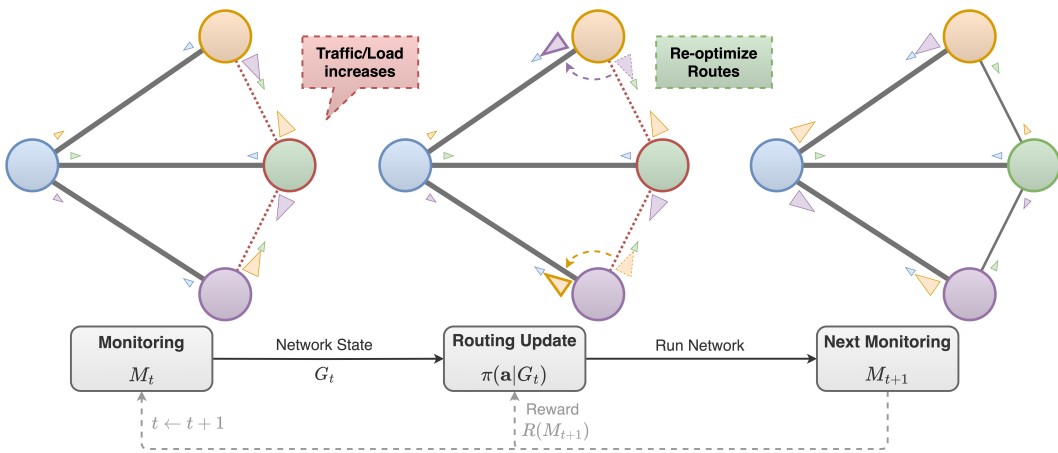

Figure 1: Situation-aware routing re-optimizes packet routes based on the network topology and current utilization and load observations to avoid congestion, delay and packet drops. Here, the longer but higher-capacity path is preferred to the shorter path when traffic spikes for the orange (top) and purple (bottom) node, causing the algorithm to re-route traffic over the blue (left) node.

As a step towards general-purpose RO, additional network information obtained via In-Band Network Telemetry (Kim et al., 2015; Tan et al., 2021) is poised to play a central yet unknown role, because the importance of individual metrics might vary between scenarios and points in time. In such scenarios, Reinforcement Learning (RL) can be used to learn a function taking as input the current network state and outputting routing decisions, by collecting data obtained via interacting with a simulator. The resulting policies are applicable in a wide range of scenarios and can improve the routing capabilities of a network compared to existing classical heuristic RPs, while requiring less manual configuration. Nevertheless, we note that related work does not fully cater to the requirements for general-purpose RO stated above (c.f. Section 2), not least because many of the approaches are evaluated on environments that oversimplify the complex dynamics of real networks. In fact, we further note that there exist no tools for extensive and reproducible evaluation for RO approaches in realistic environments.

We thus continue this important line of work by framing RL-based RO as a Swarm Markov Decision Process (SwarMDP), where a group of homogeneous agents, in this case the routing nodes in a network graph, collaborate to optimize a complex objective, in this case TEs. This formulation is the first that fulfills all requirements for general-purpose RO, and it crucially allows the resulting policies to generalize to different network topologies and load scenarios during inference. Moreover, we close the aforementioned tooling gap with *eleganTE*, a novel framework for efficiently training

and evaluating learned RO techniques for TE that leverages the ns-3 discrete-event network simulator (Henderson et al., 2008) [1]. Using this framework, we provide various benchmark scenarios that include randomly generated graphs and traffic, and showcase scenarios where OSPF and EIGRP, two of the most widely used RPs, perform subpar. Finally, we provide both a topology-dependent Multi-layer Perceptron (MLP) routing policy, and one using Graph Neural Networks (GNNs) (Veličković, 2023) that can generalize to previously unseen topologies (Section 5). While for small network topologies these policies rival the performance of OSPF and EIGRP, their results on larger networks highlight the combinatorial nature and resulting difficulty of the TE problem.[2]

## 2 RELATED WORK

**Conventional Traffic-Aware Routing Optimization.** Potential-based routing generalizes shortest-path routing by incorporating packet queue sizes into the weight computation (Basu et al., 2003). Moreover, routing configuration generation has been formulated as a constraint programming problem (Hartert et al., 2015) and extended via Local Search (LS) for sub-second re-optimization (Gay et al., 2017a). Some approaches (Jadin et al., 2019; Gay et al., 2017a) use SR as a network overlay technology for fine-grained routing control (Wu & Cui, 2023) which makes them hard to employ in new networks due to reduced compatibility (c.f. Section 1). Moreover, aforementioned SR-based TE approaches take multiple seconds or even minutes to provide a solution for larger networks ($> 50$ nodes).

**Non-Reinforcement Learning for Routing Optimization.** Some approaches attempt to learn to route without employing RL (Geyer & Carle, 2018; Rusek et al., 2022). They employ supervised learning on traffic data that corresponds to fixed routing schemes, and therefore the learned models are not guaranteed to generalize to previously unseen network situations.

**Reinforcement Learning for Routing Optimization.** Many RL approaches for RO choose to learn link weight generators for shortest-path algorithms (Stampa et al., 2017; Pham et al., 2019; Bernárdez et al., 2021; Sun et al., 2021; Chen et al., 2022; Bernárdez et al., 2023; He et al., 2023), while others use RL for direct next-hop selection (Boyan & Littman, 1993; Choi & Yeung, 1995; Ding et al., 2019; Pinyoanuntapong et al., 2019; Mai et al., 2021; Bhavanasi et al., 2022; Guo et al., 2022; You et al., 2022). Some approaches employ a middle ground, e.g. via outputting edge weights for destination-dependent next-hop forwarding forwarding (Valadarsky et al., 2017), split ratios per end-to-end communication session (Xu et al., 2018), or link weights for multi-path routing (Huang et al., 2022). Further RL approaches on RO restrict their optimization efforts to a few so-called *critical* paths (Zhang et al., 2020; Ye et al., 2022; Almasan et al., 2022), or to selection by hop count from a few candidate paths (Almasan et al., 2020).

The aforementioned approaches are insufficient with respect to the requirements stated in section 1: Many approaches lack a formalism and model architecture that permits generalization to arbitrary topologies, either because they fix input and output dimensions and thus limit the space of supported network topologies (Mai et al., 2021; Bhavanasi et al., 2022), because they rely on a particular ordering of the network state's features (Stampa et al., 2017; Valadarsky et al., 2017; Xu et al., 2018; Pham et al., 2019; Pinyoanuntapong et al., 2019; Ding et al., 2019; Sun et al., 2021; Guo et al., 2022; You et al., 2022), or because they lack a formalism altogether (Stampa et al., 2017; Valadarsky et al., 2017; Pham et al., 2019; Chen et al., 2022). Furthermore, He et al. (2023) includes the upcoming traffic demand into the formalism's state, which in our view is an unrealistic assumption, and the approaches of Bernárdez et al. (2021; 2023) need multiple model inference steps to re-optimize for a single network state, which prevents sub-second responsiveness in large networks. As these three works also lack a clear formalism on the sequential decision making nature of general-purpose RO, we indeed close a gap by providing the first clear Markov Decision Process (MDP) formulation for TE that fulfills all requirements stated in section 1.

**Tools and Frameworks.** There exist several popular datasets for network topologies (Orlowski et al., 2010; Knight et al., 2011; Spring et al., 2002), as well as random topology generators (Medina et al., 2001), of which Orlowski et al. (2010) includes traffic demands. Concerning evaluation frameworks, REPETITA (Gay et al., 2017b), which is used by Jadin et al. (2019); Bernárdez et al.

---

[1]Code and documentation will be released upon acceptance.

[2]In fact, optimal TE via link weight adjustment in link-state RPs is an NP-hard problem (Xu et al., 2011).

(2023); Almasan et al. (2022), facilitates the comparison of TE solvers by unifying the solver input and output process. A few methods evaluate on a small but real testbed network in addition to synthetic experiments (Guo et al., 2022; Huang et al., 2022), while others use custom emulator setups (Pinyoanuntapong et al., 2019; Fu et al., 2020; Huang et al., 2022) or custom simulator setups (Stampa et al., 2017; Sun et al., 2021; Chen et al., 2022; Xu et al., 2018; Pham et al., 2019). Of these, only Stampa et al. (2017) discloses its implementation, which however does not support multiple network topologies and traffic patterns.

The only publicly available training and evaluation framework for TE experiments, REPETITA (Gay et al., 2017b), assesses routing performance via computations on the abstract network graph. It does not install routing decisions on a real, emulated or simulated network, which disregards real-world interference effects caused by traffic variations, network configuration and protocol interplay. We provide a framework leveraging faithful simulation capabilities to close this gap and facilitate future research on RL for TE on a wide variety of network scenarios. Our framework also establishes the first public set of RO baselines across a diverse array of network scenarios.

## 3 DISTRIBUTED TRAFFIC ENGINEERING AS A MARKOV DECISION PROCESS

We assume that routing is single-path and unicast, meaning that for each packet at every intermediate routing node and at any point in time, there exists exactly one neighbor it is forwarded to. To formulate TE as a MDP, we split the continuous-time network operation process into time slices of length $\tau_{\text{sim}}$, after which we obtain the network state and choose the actions for the next timestep before resuming network operation.

As each node is fully defined by its current features and relation to its neighbors, we view TE as a multi-agent system of simple collaborating homogeneous agents. In contrast to existing work that employs central network views or lacks a thorough formalism (c.f. Section 2), this distributed perspective creates a robust and scalable framework for TE that works well across and generalizes to arbitrary network topologies. We consider the SwarMDP framework (Šošić et al., 2017; Hüttenrauch et al., 2019; Freymuth et al., 2023), which is a special case of decentralized partially observable MDPs designed for swarm systems, i.e., multi-agent systems with homogeneous agents. The actions represent the gateway preference choices per potential packet destination, and we extend the SwarMDP framework to variable-sized action spaces per node because nodes in computer networks may have varying numbers of neighbors. This accounts for variable agent counts in between episodes as well as permutation invariant agents, allowing for generalization to arbitrary network topologies and sizes when combined with permutation-equivariant model architectures like GNNs.

Formally, we define the TE SwarMDP as a tuple $\langle \mathbb{S}, \mathbb{O}, \mathbb{A}, T, R, \xi, \rangle$. $\mathbb{S}$ is the state space of the complete system, which in our case contains all monitoring graphs $M$ with global and local performance and load values (c.f. Sections 4, A.3). From this, the function $\xi : \mathbb{S} \rightarrow \mathbb{O}$ obtains an observation via feature selection and normalization (c.f. Section A.3), which we model as a directed graph $G_t = (V_t, E_t, \mathbb{X}_{V_t,t}, \mathbb{X}_{E_t,t}, \mathbf{x}_{u,t})$ with nodes $V_t$ and edges $E_t$ at step $t$. Node and edge features are given by $\mathbb{X}_{V_t,t} = \{\mathbf{x}_{v,t} \in \mathbb{R}^{d_{V_t}} \mid v \in V_t\}$ and $\mathbb{X}_{E_t,t} = \{\mathbf{x}_{e,t} \in \mathbb{R}^{d_{E_t}} \mid e \in E_t\}$ respectively, and $\mathbf{x}_{u,t} \in \mathbb{R}^{d_U}$ denotes optional global features. See section B.2 for further information on how $\xi$ obtains $G_t$ from $M_t$. From here, let $\mathcal{N}_u = \{v \in V_t | (u,v) \in E_t\}$ be the neighborhood of $u$ and $\Delta_k$ denote the $k$-simplex. We define the action space as a distribution over gateway preferences for each pair of current node $u$ and destination $v$ as

$$\mathbb{A} = \left\{ (u,v) \mapsto \mathcal{D}_{u,v} \mid u,v \in V, \mathcal{D}_{u,v} \in \Delta_{|\mathcal{N}_u|-1} \right\}. \tag{1}$$

In other words, each routing node $v \in V$ represents an agent that specifies a distribution over its neighbors per possible destination node. This preserves the homogeneous nature of agents while allowing for varying neighborhood sizes, since the agents specify their routing preferences in the same way. The transition function $T : \mathbb{S} \times \mathbb{A} \rightarrow \mathbb{S}$ is unknown in practice, since the decision model does not have access to the upcoming traffic demands. $R : \mathbb{S} \times \mathbb{A} \rightarrow \mathbb{R}$ is a global reward function.

Related work has optimized for multiple commonly used performance markers like maximizing throughput (Fu et al., 2020), or minimizing maximum link utilization (Bernárdez et al., 2023; Chen et al., 2022), packet delay/latency Guo et al. (2022); Sun et al. (2021) or drop counts (Fu et al., 2020). In our experiments, we use the composite reward function $R(\mathbf{s}_t, \mathbf{a}_t) = -(\rho_{\text{wd}} R^{\text{wd}} + \rho_{\text{dr}} R^{\text{dr}})$,

where

$$R^{\mathrm{wd}}(\mathbf{s}_t, \mathbf{a}_t) = \frac{1}{|P_t^{(+)}| + |P_t^{(-)}|} \left( \sum_{p \in P_t^{(+)}} d(p) + \lambda_{P^{(-)}} d_t^{\max} |P_t^{(-)}| \right)$$

is the *weighted delay* of packets in which each dropped packet $p \in P_t^{(-)}$ is penalized with the maximum delay $d_t^{\max}$ that occurred in timestep $t$ weighted by $\lambda_{P^{(-)}}$, while received packets $p \in P_t^{(+)}$ are penalized with their delay values $d(p)$, and

$$R^{\mathrm{dr}}(\mathbf{s}_t, \mathbf{a}_t) = \frac{|P_t^{(-)}|}{|P_t^{(+)}| + |P_t^{(-)}|}$$

is the *drop ratio* at timestep $t$. The values used in our experiments for the hyperparameters $\rho_{\mathrm{wd}}$, $\rho_{\mathrm{dr}}$ and $\lambda_{P^{(-)}}$ can be found in Section B.4. Also, see Section D.2 for ablations on reward functions.

Our goal is to find a policy $\pi : \mathbb{S} \times \mathbb{A} \to [0, 1]$ that maximizes the return, i.e., the expected discounted cumulative future reward $J^t := \mathbb{E}_{\pi(\mathbf{a}|\mathbf{s})} \left[ \sum_{k=0}^{\infty} \gamma^k R(\mathbf{s}^{t+k}, \mathbf{a}^{t+k}) \right]$. Here, an optimal policy jointly minimizes the weighted delay of packets and their drop ratio over the course of a simulation depending on the current monitoring graph. Note that this is different from the way that heuristics like OSPF usually do routing, in that there is a principled objective that is optimized.

# 4 A WORKBENCH FOR TRAFFIC ENGINEERING EXPERIMENTS

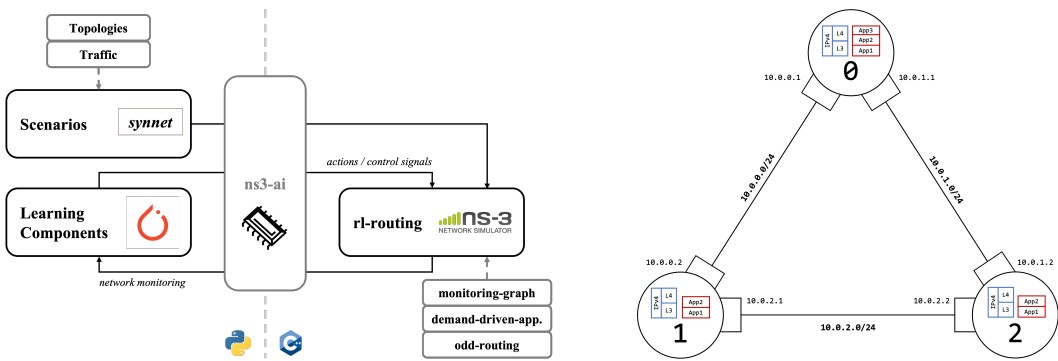

Figure 2: Left: Structural overview of eleganTE. Right: Example 3-node network setup in ns-3 incl. applications (red boxes) and Internet Stack (blue boxes).

*eleganTE* is an RL framework that interfaces the discrete-event network simulator ns-3 (Henderson et al., 2008) for repeatable and highly configurable RO experiments with realistic network models (see Section A.1 for details on simulation in ns-3). The *rl-routing* component is eleganTE's core component and implements the TE SwarMDP of Section 3 (see Section A.2 for further details). It uses the shared memory module of ns3-ai (Yin et al., 2020) to facilitate communication between learning algorithm, data generator and simulation modules. It uses three custom ns-3 extension modules to obtain the required simulation and telemetry capabilities:

- The *monitoring-graph* module provides the network monitoring graphs $M_t$ via in-band telemetry (Kim et al., 2015). In monitoring graphs, each Point-to-Point (P2P) connection between nodes $u$ and $v$ is modeled as two directed edges $(u, v)$ and $(v, u)$ to incorporate utilization statistics of the respective sender network device into the edge. See Section A.3 for further details on how the values for $M_t$ are obtained.

- The *demand-driven-application* module provides source and sink applications that can be filled with new Traffic Matrix (TM) demands at the start of each timestep. Each source application then employs a constant-bitrate sending rate so that they, over the course of $\tau_{\mathrm{sim}}$, send data volumes corresponding to the TM entries to the specified receivers. By default data is sent as UDP traffic, however at installation time each source application is converted into a TCP sender with a configurable simulation-wide probability of $p_{\mathrm{TCP}}$.

- The *odd-routing* module is a drop-in replacement for other routing protocols like OSPF that allows the installation of routing actions $A_t \in \mathbb{A}$ onto the network nodes (see Section A.4 for further details on action installation).

## 4.1 VERSATILE NETWORK TOPOLOGIES AND TRAFFIC MATRICES

We strive for versatile simulation conditions in our framework and have created *synnet* as a standalone module that provides suitable *network scenarios* for our purpose. Network scenarios consist of the network topology (i.e. the routing nodes and links between them, as well as parameters such as link datarate and delay) and traffic dynamics over the course of the episode, modeled as a TM per timestep. We currently support both a small range of pre-defined topologies (Figure 8), as well as random topology graphs of arbitrary node counts (Figures 9 and 10). We use the gravity model (Roughan, 2005) to generate TMs, scale them by timestep-dependent coefficients and introduce small random perturbations to increase the variety of covered traffic dynamics. See Section A.5 for further details on scenario generation.

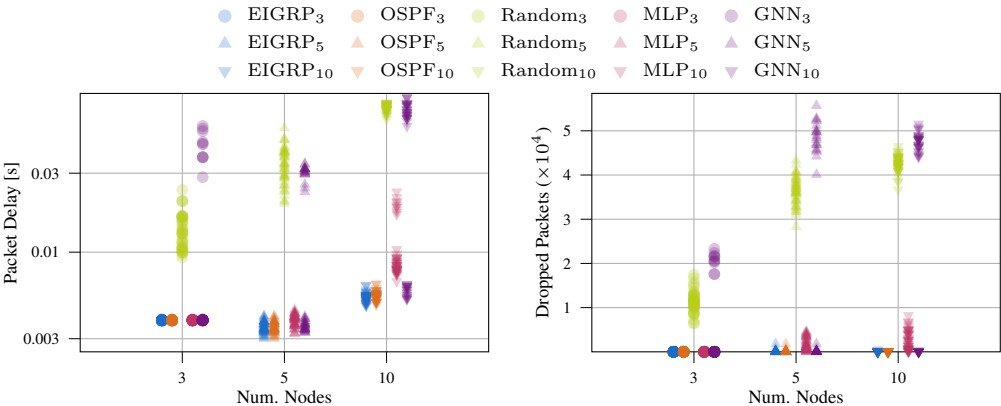

Figure 3: Performance statistics for the three topologies *predef3*, *predef5* and *predef10*. Each data point represents an evaluation. Left: Packet delay values per approach for 3, 5 and 10 nodes. Right: Dropped packet counts per approach for 3, 5 and 10 nodes.

## 5 EXPERIMENTS

## 5.1 LEARNED ROUTING OPTIMIZATION

In the following, we discuss and design policies that are compatible with the action space of Equation 1. In general, each policy consists of an *actor* module and an *assignment* module. The actor module $\phi : \mathbb{O} \rightarrow \mathbb{R}^{|V| \times |E|}$ provides an unnormalized value for each combination of destinations and gateways given the current monitoring graph. Intuitively, this module describes how well each graph edge is suited as a next-hop route for packets with a given destination. The *assignment* module $\psi : E \times \mathbb{R}^{|V| \times |E|} \rightarrow \mathbb{A}$ then maps these values to gateway probabilities. To efficiently explore the action space, we model the actor component as an isotropic Gaussian $\phi(\mathbf{s}) = \mathcal{N}(\mu(\mathbf{s}), \mathbf{I}\sigma)$ for a learnable standard deviation $\sigma$. We compare this to other exploration schemes in Appendix D.3.

Since the nodes of the network graph are permutation invariant, one option is to parameterize the actor module as a GNN (Bronstein et al., 2021; Veličković, 2023). Here, we use Message Passing Networks (MPNs) Sanchez-Gonzalez et al. (2020); Pfaff et al. (2021); Linkerhägner et al. (2023) as they are the most general form of GNNs (Bronstein et al., 2021). Our MPN consists of *L Message Passing Steps*, where each step $l$ updates latent node and edge features of a given graph using information from the previous step. Using MLPs $f^l$ and initial node and edge features $\mathbf{x}_v^0$ and $\mathbf{x}_e^0$, the $l$-th step is given as

$$\mathbf{x}_e^{l+1} = f_E^l(\mathbf{x}_v^l, \mathbf{x}_u^l, \mathbf{x}_e^l),$$

$$\mathbf{x}_v^{l+1} = f_V^l\left(\mathbf{x}_v^l, \frac{1}{|\{(v,u)\}|} \sum_{e=(v,u)} \mathbf{x}_e^{l+1}\right), \text{ with } e = (v,u) \in E.$$

The network's final output is a learned representation $\mathbf{x}_v^L$ for each node $v \in \mathcal{V}$ that encodes information about the graph topology and the current local monitoring state of each node and edge. As this encoding does not provide information about pairs of gateways and destinations, we combine it with an auxiliary distance measure over pairs of nodes and feed both into a readout layer that is shared across edges to yield the desired action. Crucially, this parameterization is independent of the topology and size of the network graph, allowing a single learned policy to generalize to novel topologies during inference.

As an alternative, we model the actor module as a simple MLP that gets as input a concatenated vector of the current of the current observation. This variant fully relies on a fixed ordering and size of the nodes and edges, and thus cannot generalize beyond the specific network topology graph it is designed for and trained on. Additional details for the modules are given in Section B.5. We train the policies using Proximal Policy Optimization (PPO) (Schulman et al., 2017) with scalar continuous actions per edge. See Section B.4 for PPO hyperparameter details.

We compare these policies to classical RO heuristics, namely OSPF (Moy, 1997) and EIGRP (Savage et al., 2016). Both use standard reference values for datarate and delay as detailed in Section B.3. Additionally, we provide an untrained random policy for reference.

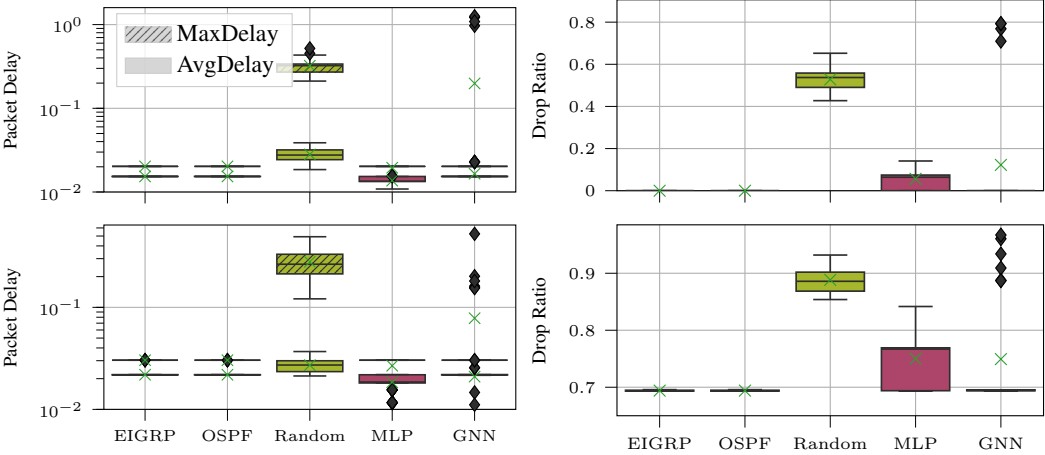

Figure 4: Evaluation results on *predef4s* in *flat* (top) and *peak* (bottom) traffic mode. Left: Average (full boxes) and maximum (shaded boxes) packet delay. Right: ratio of dropped to sent packets.

## 5.2 BENCHMARKS

We define a non-exhaustive list of benchmarks using eleganTE in increasing order of difficulty (c.f. Figures 8, 9, 10). All benchmarks consider network topologies that stay constant within an episode, and unless mentioned otherwise we use the *peak* traffic mode which invariably introduces short periods of fully utilized links (i.e. the maximum Link Utilization (LU) per episode is always 1).

- *predef3s* and *predef4s* are predefined topologies designed to highlight the shortcomings of shortest-path algorithms. Only nodes 0 and 1 are equipped with sending and receiving applications.

- *predef3*, *predef5* and *predef10* are predefined topologies of 3, 5, or 10 nodes. Each node is equipped with sending and receiving applications which results in dense generated TMs.

- The *nxN* series consists of randomly generated topologies with arbitrary node counts N. We report evaluation results on *nx10*, *nx25* and *nx50*.

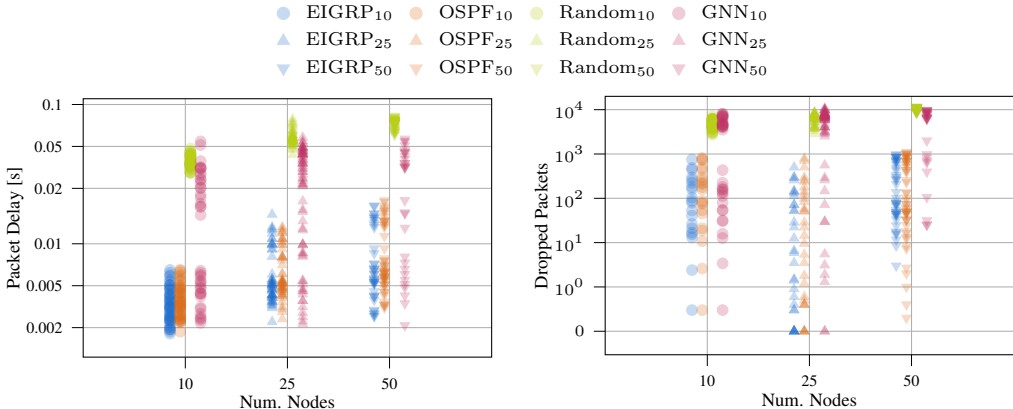

Figure 5: Performance statistics for random topologies of 10, 25 and 50 nodes for different baseline heuristics and a GNN policy trained on random topologies with 10 nodes. Each data point represents an evaluation. Left: Average packet delay values per approach for 10, 25 and 50 nodes. Right: Dropped packet counts per approach for 10, 25 and 50 nodes.

## 6 RESULTS

We train all learned methods for 20000 episodes and use the hyperparameters of Section B unless mentioned otherwise. The episode length is $T = 32$, consisting of 24 TMs generated with synnet followed by 8 timesteps in which no new traffic is ingested into the network. This results in 640000 training steps, equaling roughly 17.8 hours of simulated network time given $\tau_{\text{sim}}$, and 1250 training iterations. Each model is trained on 5 cores of an Intel Xeon Platinum 8358 CPU for up to two days. We evaluate the trained policies and baselines on 10 randomly generated evaluation episodes. We report the performance of the tested approaches based on the optimization criteria mentioned in Section 3, showing minimum, maximum and interquartile means of the evaluated metrics across 5 random seeds (Agarwal et al., 2021) for each method.

### 6.1 PREDEFINED NETWORK TOPOLOGIES

Figure 3 shows results on the predefined topologies *predef3*, *predef5* and *predef10*. The learned policies geberally perform on par with OSPF and EIGRP for most episodes, but are overall less stable. The performance for learned methods degrades with increasing network size, indicating a more complex learning problem for larger network graphs. Further, we provide results for the *predef4s* scenario in Figure 4. The learned policies are able to achieve lower delay values than both OSPF and EIGRP in some cases for both low and high traffic scenarios, showing the potential of learned TE compared to traditional methods. The GNN policy however exhibits a very high variance in its results, hinting at unstable convergence properties.

### 6.2 RANDOM NETWORK TOPOLOGIES

Figure 6 shows the performance of the baselines as well as the GNN policy on random network topologies of 10 nodes. While the GNN policy achieves strong performance on several topologies, we again note a very high variance. We further evaluate the generalization capabilities of our GNN policy. Section C shows the performance on random 10 node graphs when trained only on *predef10*. Furthermore, Figure 5 highlights the generalization capabilities of the proposed GNN policy to larger networks.

### 6.3 TCP TRAFFIC

We report results on the *predef5* topology when sending TCP traffic instead of UDP, which is characterized by the additional congestion control mechanism that throttles the application sending rate if packet queues build up. Moreover packet drops are much more severe since data packets have

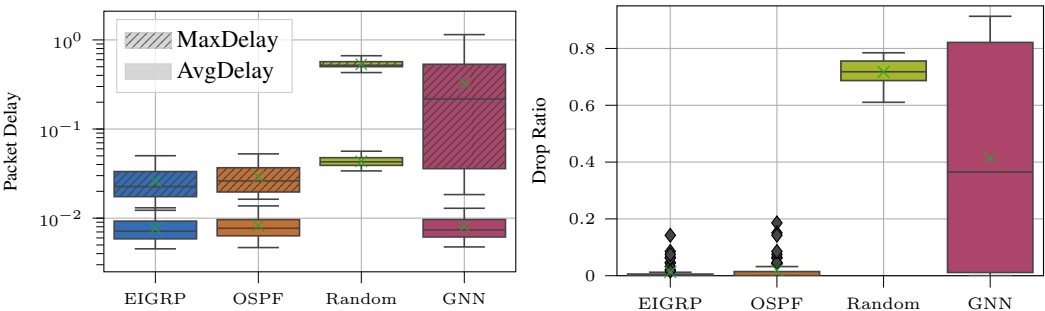

Figure 6: Results for random topologies of 10 nodes. Left: Average (full boxes) and maximum (shaded boxes) packet delay. Right: ratio of dropped to sent packets.

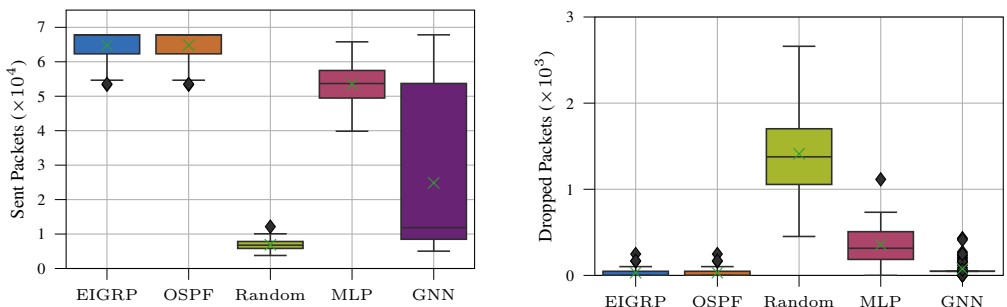

Figure 7: Results per episode for experiments on *predef5* with TCP traffic. Left: Sent packets. Right: Dropped packet counts.

to be recieved in order. We can thus assess RP performance via the amount of sent and dropped packets. Figure 7 shows that both policies can perform on par with the baseline RPs, although not reliably.

## 7 CONCLUSION

We formulate distributed TE as a SwarMDP and thus provide the first formalism that fulfills all requirements for general-purpose RO as stated in Section 1. Also, we provide the implementation of this formalism via *eleganTE*, a flexible and powerful framework for training and evaluating RL agents for realistic RO settings. As shown by the range of presented experiments, eleganTE facilitates repeatable experiments on network scenarios with a large variety in topology and traffic patterns. Our presented policies rival popular shortest-path RPs in many scenarios and expose their weaknesses, but also leave room for improvement concerning performance and stability. This motivates the need for and potential of further research on RL algorithms for general-purpose ROs.

### 7.1 LIMITATIONS AND FUTURE WORK

While our framework eleganTE and the policies presented in Section 5 cater to the requirements of Section 1, we leave the evaluation of scenarios with changing topologies and corresponding policies for future work. The training stability of our policies is an issue that we partly attribute to the stability issues PPO is known for (Engstrom et al., 2020), but it is conceivable that adjustments policy and hyperparameters bring further performance improvements. Furthermore, for truly distributed TE a decentralized training and execution paradigm is necessary and requires adjustments to the presented policies as well as novel approaches for router information exchange. Finally, the usefulness of eleganTE as a training and evaluation platform can be further extended via experiments on real traffic and topology data (Orlowski et al., 2010; Knight et al., 2011; Spring et al., 2002), or by adding support for traffic flows and flow completion times to our traffic generation and evaluation capabilities (Dukkipati & McKeown, 2006).

## ETHICS STATEMENT

Distributed TE with RL is an essential step towards automating computer networks, which can greatly increase operational efficiency and save costs through over-provisioning or manual configuration. Other domains such as transport networks or power grids might benefit from progress in distributed TE too, due to their structural similarity. Our work opens the door for future research on this exciting and challenging research problem via the proposed training and evaluation framework and the presented policies. While we highlight the value and potential benefits of our research, like most research on RL misuse for malicious intents is conceivable. Specifically for RL approaches in computer networks, the black-box nature of policy architectures can potentially be used to infiltrate the network's decision making, causing disturbances if appropriate security measures are not taken. Furthermore, it is conceivable that different traffic demands are not treated equally by the routing policy, putting certain kinds of traffic at an unnatural disadvantage. Concerning the legal aspects of our work, we do not use any proprietary datasets or code for our work, nor do we collect or process personal data. No affiliations or financial interests exist that might compromise the objectivity and integrity of this work.

## REPRODUCIBILITY STATEMENT

In order to facilitate the replication of our experiments and the validation of our results, we will provide the complete source code as well as extensive documentation upon acceptance. Furthermore, we provide a comprehensive description of the experimental setup, proposed framework, hyperparameters, data generation, training and evaluation procedures in Sections 4 to 6 of the main paper, and Sections A to D of the appendix. We also provide a clear explanation of the proposed SwarMDP framework in Section 3.

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

# A ELEGANTE: FRAMEWORK DETAILS

## A.1 NETWORK SIMULATIONS IN NS-3

Networks in ns-3, by default, consist of nodes and links/connections between nodes (see Figure 2, right side). For modeling simplicity, we limit ourselves to full-duplex P2P connections that transmit data error-free and at a constant pre-specified datarate. Nodes themselves do not generate or consume data; Instead, applications are installed on nodes that generate data destined for other applications, or consume the data that is destined for them (red boxes in example network nodes in Figure 2). To transport data between nodes we install an *Internet Stack* on top of each node, adding IP and TCP/UDP components in a way that mimics the OSI reference model (Zimmermann, 1980). Also, nodes do not put data on the P2P link themselves, or read data from it. This is done by the network devices that belong to a P2P connection, which are installed as interfaces on the two nodes that are being connected. Upon installation of the Internet Stack, the P2P connection between two nodes is assigned an IP address space, with concrete IP addresses given to the incident network devices. After topology creation and application installation, a *Simulator* is started: the installed source applications send data to the specified destination nodes as configured, which passes the OSI layers as usual and gets wrapped into IP packets as they enter the routing plane. The RP that has been installed with the Internet Stack fills each node's routing table and performs lookups when outgoing or incoming IP packets arrive, forwarding these packets to the specified next-hop neighbor or locally delivering them to the sink applications.

## A.2 IMPLEMENTING THE TE SWARMDP

At the start of a new episode, the learning loop starts a new rl-routing instance in a subprocess and provides a network scenario generated with synnet (Sections 4.1, A.5) which comprises the network topology as well as a series of TMs that cause data traffic at each time step. rl-routing first installs the contained network topology in ns-3 and configures nodes, links and network devices accordingly. Thereafter, each node installs demand-driven source and sink applications. The initial state $S_0$ is converted from of an initial monitoring $M_0$ (Section A.3). The learning side processes the current monitoring and communicates a routing action $A_t$ to rl-routing alongside the upcoming traffic requirements. The routing actions are installed as described in Section A.4, then rl-routing simulates the installed network for a duration of $\tau_{\text{sim}}$ and pauses the simulation. The network monitoring results $M_{t+1}$ for timestep $t$ are used to calculate $r_t$ and converted into the new state $S_{t+1}$. After rl-routing has simulated the last provided TM in timestep $T$, the learning loop sends a termination signal to the rl-routing subprocess which in turn terminates the simulation.

## A.3 PACKET MONITORING IN NS-3

The *monitoring-graph* module keeps track of the network performance during simulation. After each timestep as well as after start of a new episode, it builds a clone of the network topology graph where, as opposed to the undirected topology graph, each P2P connection between nodes $u$ and $v$ is modeled as two directed edges $(u, v)$ and $(v, u)$ to incorporate utilization statistics of the respective transmitter network device into the edge. In order to acquire detailed utilization information from the simulation steps, we trace the circulating packets and log the following event types during simulation:

- E1: packets leaving sender applications/arriving at sink applications,
- E2: packets getting enqueued/dequeued in network device buffers,
- E3: packets getting dropped (incl. drop reason),
- E4: packets getting passed downward to the routing layer at the sender node (once per packet),
- E5: packets getting passed upward from the routing layer for local delivery at the receiver node (up to once per packet).
- E6: packets getting put on a P2P connection from the outbound network device for transmission/read from a P2P connection from the inbound network device.

Using these event categories, we calculate the following performance metrics and store them in the directed monitoring graph:

- The amount of sent and received packets (globally and per node) per timestep is calculated from events of type E1.

- For each network device, its buffer load is tracked using events of type E2 and the maximum and latest buffer load values are stored at the end of a timestep.

- The amount of globally dropped packets for the current timestep is obtained using events of type E3.

- By packet delay, we denote the total routing delay between the first time the packet enters the IP layer, and the time it leaves it for local delivery (obtained from events of type E4 and E5). We track the global average and maximum packet delay per timestep.

- The amount of sent/received/dropped packets per P2P connection per timestep is obtained from events of type E3 and E6.

At the end of a timestep $t$, $M_t$ holds global, node and edge features that reflect the overall network performance and utilization during timestep $t$, as well as its load state at the end of timestep $t$. For the initial monitoring $M_0$, these values are set to zero.

## A.4    ON-DEMAND DISTRIBUTED ROUTING IN NS-3

As indicated in section 3, for each routing node in the network we communicate a vector of next-hop neighbor selections per conceivable destination node. The *odd-routing* module closely resembles the other IPv4 RP modules implemented in ns-3, leveraging the line-speed capability of the forwarding plane by using routing table lookups and thus fulfilling the "compatibility" requirement stated in section 1. But since usually not all node pairs in a network communicate with each other, it stores the received routing actions in a separate location on the routing node, and only fills the node routing table on-demand once a packet arrives for which no suitable routing rule is found in the routing table. All subsequent packets destined for the same target node will have access to the newly installed table entry until the start of a new timestep, when new routing actions will be stored in the node and its routing table will be flushed.

## A.5    SCENARIO GENERATION

Network topologies vary greatly depending on the scope and use case of the network. For this work, we orientate our scenario generation towards the topologies spanned by the edge routers that connect datacenters in typical Inter-Datacenter Wide Area Networks (Inter-DC WANs). These are usually characterized by loosely meshed powerful edge routers and high-datarate medium-latency links that connect two datacenters each. While we also employ link delay values in the low ms range, we scale down typical datarate values for Inter-DC WANs to lie in the high Mbps range, to speed up simulation times under stress situations without loss of generality of the simulation results. For simplicity, we set the packet buffer sizes of network devices incident to P2P connections to the product of link datarate and delay, which is common throughout the networking literature (Spang et al., 2022).

Figure 8 shows the pre-defined topologies used for our experiments. To generate random network topology graphs, we use the BA (Barabási & Albert, 1999), the ER model (Erdős et al., 1960) and the WS (Watts & Strogatz, 1998), all available via NetworkX graph analysis package (Hagberg et al., 2008). Figures 9 and 10 show examples for such random topology graphs. In any case, nodes and edges are assigned unique integer IDs for identification purposes. To add the missing datarate and delay values to the links, we follow the following steps:

- We first embed the random graph into a two-dimensional plane using the Fruchterman-Reingold force-directed algorithm (Fruchterman & Reingold, 1991) to create synthetic positional information for the random graph's nodes, similar to the position information provided for nodes in related network datasets (Orlowski et al., 2010; Knight et al., 2011; Spring et al., 2002).

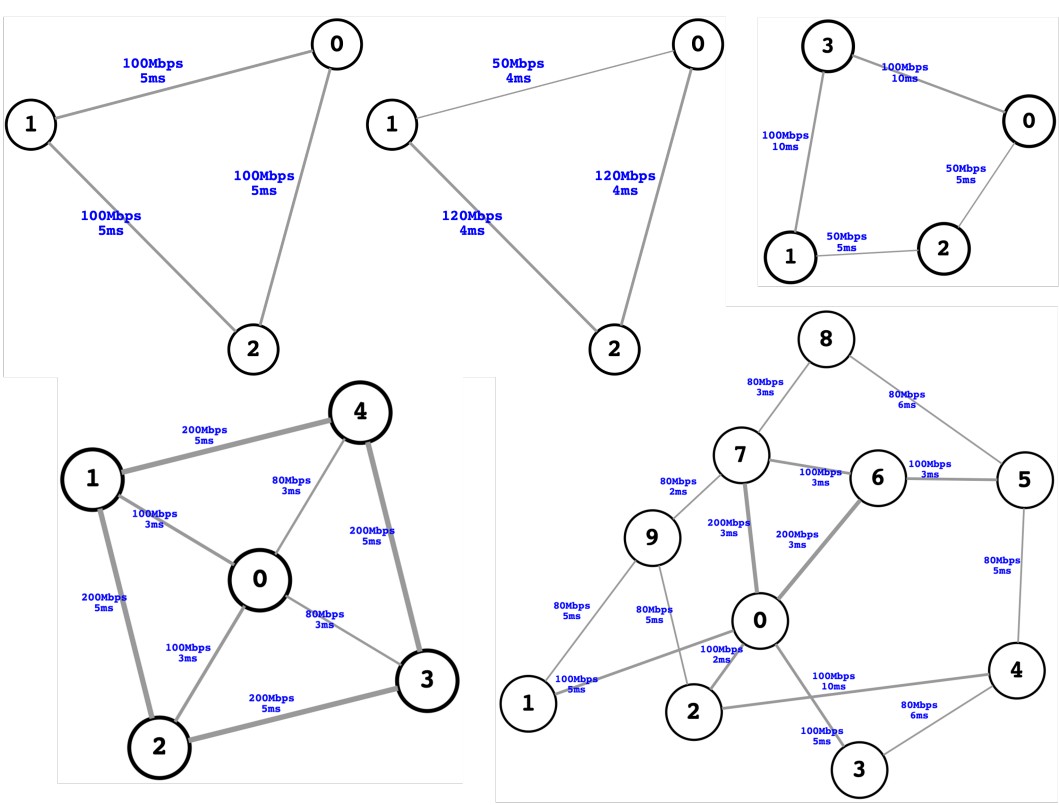

Figure 8: Pre-defined network topologies used for this paper. Upper row from left to right: *predef3*, *predef3s*, *predef4s*; Lower row: *predef5* and *predef10*
.

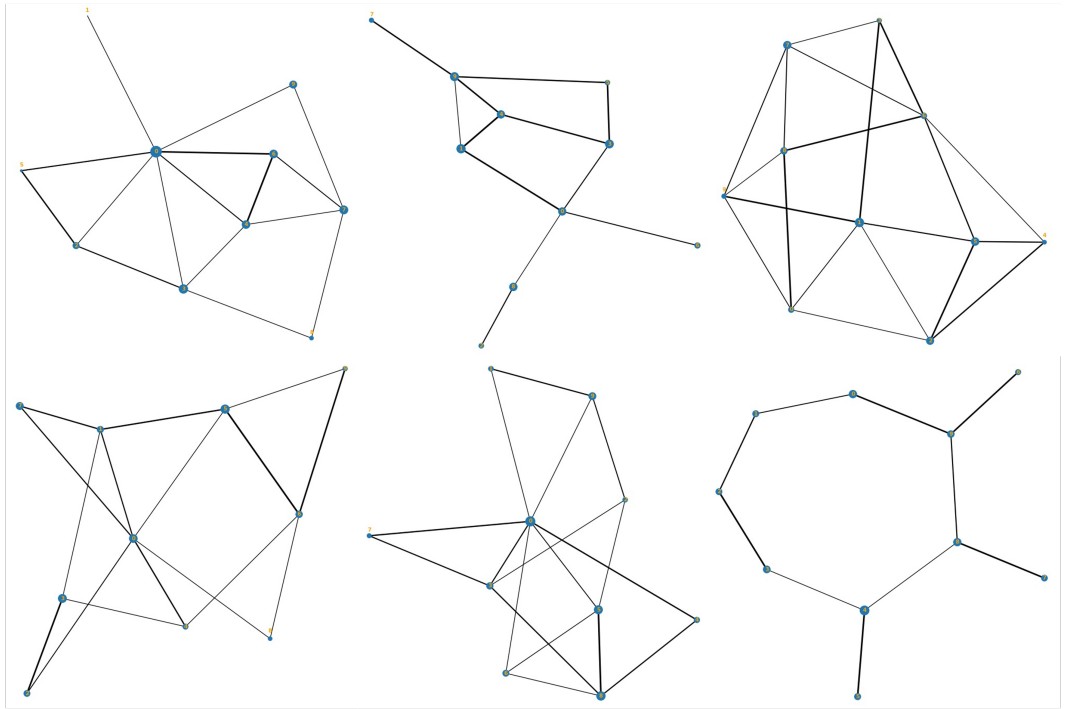

Figure 9: Examples of 10-node network topologies generated with NetworkX. Bigger nodes indicate higher node weights, thicker edges indicate higher edge weights. Columns from left to right (2 examples each): Barabási-Albert (BA), Erdős-Rényi (ER), Watts-Strogatz (WS).

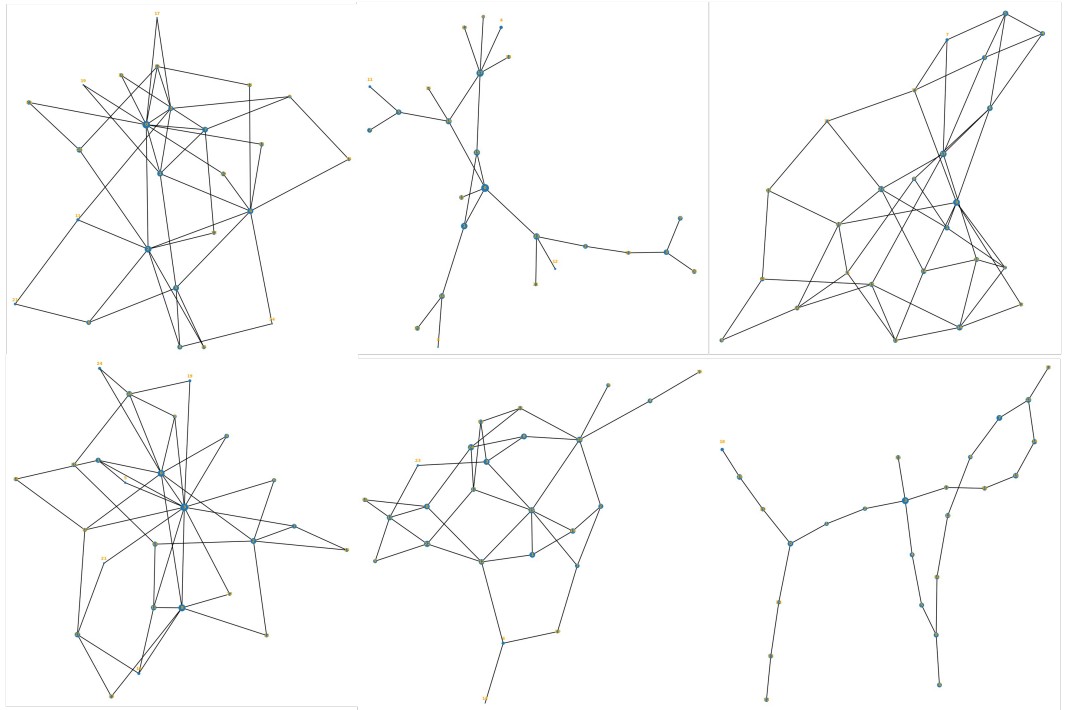

Figure 10: Examples of 25-node network topologies generated with NetworkX. Bigger nodes indicate higher node weights, thicker edges indicate higher edge weights. Columns from left to right (2 examples each): BA, ER, WS.

- The resulting positional layout is centered around the two-dimensional point of origin, which we use to obtain location weights per node that are inversely proportional to its distance to the origin. We scale the location weight of each node with its degree and use the scaled and normalized weights to obtain a node weight between in the pre-specified interval $[c_{\min}, c_{\max}]$.
- The node weights are randomly perturbed by factors between $2^{-\delta_{\text{node}}}$ and $2^{\delta_{\text{node}}}$.
- Next we set the edge weights, which are used to obtain the link datarates, to be the maximum of the corresponding incident nodes' weights. We rescale them to lie in the pre-specified interval of minimum and maximum datarates $[v_{\min}, v_{\max}]$.
- We obtain delay weights per edge from the euclidean distance of the incident nodes' embeddings and normalize them so that the average edge weight is equal to a pre-specified value $v_{\text{mean}}$.
- The edge datarate and delay values are randomly perturbed by factors between $2^{-\delta_{\text{edge}}}$ and $2^{\delta_{\text{edge}}}$, ensuring that two successively generated scenarios will not be the same.

In order to generate different types of traffic for both predefined and randomly generated network graphs, we first specify different modes of traffic intensity progression. They are implemented by calculating a fill coefficient $f_t$ per timestep $t$ which scales the generated TMs.

- The *flat* traffic mode aims at keeping the average traffic demand at a constant value by setting $f_t$ to a pre-defined constant value $f_{\text{flat}}$.
- The *peak* traffic mode models a single spike in traffic values and is designed as a stress-test scenario that invariably introduces some amount of congestion. For $T$ timesteps, starting from $f_{\min}$ for $t = 0$, the coefficient linearly increases to $f_{\max}$ at $t = \frac{T}{2}$ and linearly decreases back to $f_{\min}$ for $t = T$.

Finally, the TMs are generated according to the gravity model (Roughan, 2005):

$$\text{TM}_t = \texttt{v\_mean(G)} \cdot f_t \cdot \tau_{\text{sim}} \cdot \texttt{a\_hc(G)} \cdot \texttt{n\_ac(G)}^2 \cdot \mathbf{p}_{\text{in}} \mathbf{p}_{\text{out}}^T \tag{2}$$

where $\texttt{v\_mean(G)}$ denotes the average datarate of edges in $G$, $f_t$ denotes the flow fill coefficient at timestep $t$, $\texttt{a\_hc(G)}$ denotes the average hop count between nodes in $G$, and $\texttt{n\_ac(G)}$ denotes the number of active sender/receiver nodes in $G$. The vector $\mathbf{c}$ of node weights $c_i$ is used for both $\mathbf{p}_{\text{in}}$ and $\mathbf{p}_{\text{out}}$ in the TM generation of the gravity model.

## B  HYPERPARAMETERS AND DEFAULTS

The listed default hyperparameters and settings are used in all our experiments unless mentioned otherwise.

### B.1  SIMULATION IN NS-3

We set up the applications to send data packets of 1472 bytes, which accounts for the commonly used IP packet maximum transmission unit of 1500 bytes and the sizes for the IP (20 bytes) and ICMP (8 bytes) packet header. UDP packets thus are 1500 bytes large, whereas TCP may split up data units received from the upper layer as required. We set the simulation step duration $\tau_{sim}$ to 100ms and $p_{\text{TCP}} = 0$, meaning that by default we experiment on UDP traffic only (see section 6.3 for experiments on traffic that is partly or fully TCP).

### B.2  MONITORING FEATURES

We implement the observation function $\xi$ by using the values of monitoring graph $M_t$ (Section A.3) as features in its corresponding state graph $G_t$ as follows:

- **global:** none

- **node:** global maximum link utilization (`maxLU` $\in [0,1]$), global average datarate utilization `avgTDU` $\in [0,1]$, global average packet delay `avgPacketDelay` $\in \mathbb{R}^+$ and global maximum packet delay `maxPacketDelay` $\in \mathbb{R}^+$.

- **edge ($\phi_{\text{GNN}}$):** link utilization `LU` $\in [0,1]$, maximum relative packet buffer fill `txQueueMaxLoad` $\in [0,1]$, relative packet buffer fill at end of simulation step `txQueueLastLoad` $\in [0,1]$.

- **edge ($\phi_{\text{MLP}}$):** all the above edge features, plus packet buffer capacity (relative to the maximum packet buffer capacity of all edges) `capacity` $\in [0,1]$ and channel delay (relative to the highest delay values of all edges) `delay` $\in [0,1]$.

Consequently, we have $d_U = 0$, $d_V = 4$, $d_{E,\text{GNN}} = 3$ and $d_{E,\text{MLP}} = 5$.

### B.3 OSPF AND EIGRP WEIGHT CALCULATION

The default calculation formula for OSPF link weights is

$$\text{weight}(e) = \frac{v_{\text{ref}}^{\text{OSPF}}}{v(e)}$$

where $v(e)$ denotes the datarate value of link $e$ and the reference datarate value $v_{\text{ref}}^{\text{OSPF}}$ is set to $10^8$ (Moy, 1997). We use the classic formulation for EIGRP link weights with default K-values, which yields

$$\text{weight}(e) = 256 * \left( \frac{v_{\text{ref}}^{\text{EIGRP}}}{v(e)} + \frac{d(e)}{d_{\text{ref}}^{\text{EIGRP}}} \right)$$

where $d(e)$ denote the delay value of link $e$, the reference datarate value $v_{\text{ref}}^{\text{EIGRP}}$ is set to $10^7$ and the reference delay value $d_{\text{ref}}^{\text{EIGRP}}$ is set to 10 (Savage et al., 2016).

### B.4 PPO

Each training iteration uses 512 sampled environment transitions to do 10 update epochs with a minibatch size of 128. We multiply the value loss function with a factor of 0.5, clip the gradient norm to 0.5 and use policy and value clip ratios of 0.2 as per Schulman et al. (2018). We use a discount factor of $\gamma = 0.97$ and use $\lambda_{\text{GAE}} = 0.95$ for Generalized Advantage Estimation (Andrychowicz et al., 2020). We set the reward scaling factors $\rho_{\text{dr}} = 0.2$, $\rho_{\text{loop}} = 1$, $\rho_{\text{LU}} = 0.2$, $\rho_{\text{wd}} = \rho_{\text{ad}} = \rho_{\text{md}} = 10$ and $\lambda_{P(-)} = 2$. These scaling factors have been selected so that the expected average values for the individual scaled reward components lie in the same order of magnitude (and thus each corresponding objective is valued roughly equally), except for $R_{\text{loop}}$ which can range considerably higher the more loops have been introduced to the routing. We model the value function baseline that PPO uses for variance reduction as separate network that is defined analogous to the respective policy, but uses a mean over all outputs to provide a value estimate of the global state.

### B.5 POLICY IMPLEMENTATION

We implement our Neural Network (NN) modules in PyTorch (Paszke et al., 2019) and use the Adam optimizer with a learning rate of $\alpha = 1e\text{-}4$ (Kingma & Ba, 2014). For our MLPs we use 2 layers with a latent dimension of 32, and likewise use 2 message passing layers with a latent dimension of 32 in our MPNs. Both modules use *LeakyReLU* activation functions. We apply layer normalization (Ba et al., 2016) and residual connections (He et al., 2016) to node and edge features independently after each message passing step. The actor's standard deviation is parameterized as $\sigma = e^{\varepsilon}$, where $\varepsilon$ is initially set to $-1$ and learned alongside the other policy parameters. Concerning the assignment module $\psi$, we choose the $\arg\max$ operator to obtain gateway preferences: $A_i = \langle \arg\max_{e=(u,v)} A'_{ej} \mid j \in V, v \in \mathcal{N}_i \rangle$. In our experiments, the auxiliary distance measure provided to the readout of the GNN actor module is the sum of EIGRP link weights for the shortest path from $i$ to $j$.

## B.6    Scenario Generation

For the BA and ER models we choose the attachment count $m \in \{2, 4\}$ uniformly, for the rewiring probability in the WS model we choose $p_{\text{rewire}} \in \{0.2, 0.3, 0.4\}$ uniformly and for the ER model we choose an average node degree of $\deg_{\text{avg}} \in \{2, 3, 4\}$ uniformly and use it to set an according edge creation probability.

Concerning traffic generation, we use node weight interval borders of $c_{\min} = 50 \cdot 10^6$ and $c_{\max} = 200 \cdot 10^6$, node weight perturbation factors of $\delta_{\text{node}} = 1$. For the edge weights we use interval borders of $v_{\min} = 50 \cdot 10^6$ and $v_{\max} = 200 \cdot 10^6$, an average weight $v_{\text{mean}} = 100 \cdot 10^6$ and an edge weight perturbation factor of $\delta_{\text{edge}} = 0.5$. We we use the *peak* traffic mode and set the traffic fill coefficients to $f_{\min} = 0.5$ and $f_{\max} = 5.0$.

## C    Additional Results

### C.1    Generalizing from a Single Training Topology

Figures 11 shows evaluation results on random graphs of 10 nodes, where the GNN policy has been trained only on the pre-defined topology *predef10*. While the learned policy performs favorably for a small set of random topologies and even achieves lower delay values then OSPF and EIGRP on some topologies, its performance collapses on others, indicating that training on a versatile of network topologies is crucial for performance.

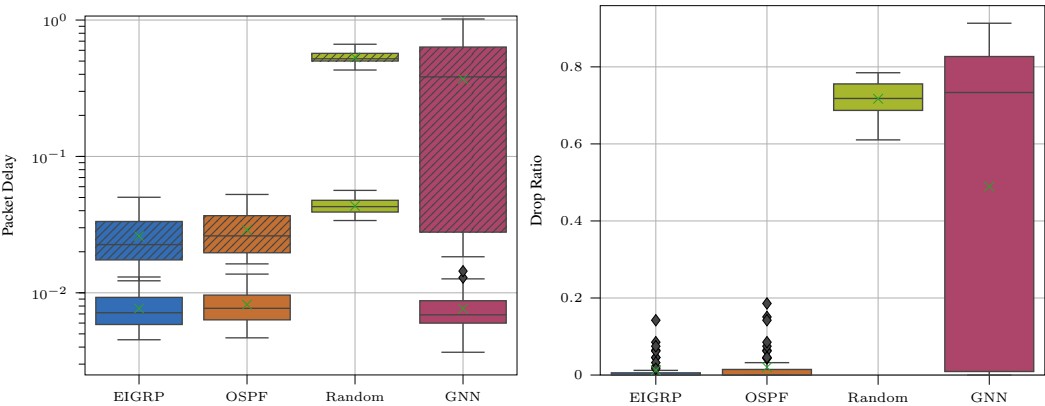

Figure 11: Average and maximum packet delay (left) and dropped packet ratio (right) per evaluation episode on random graphs of 10 nodes, where the policies have been trained only on the pre-defined topology *predef10*.

### C.2    Route choice

Figure 12 shows the results for training runs on *predef3s* using both the *peak* traffic mode as well as the *flat* traffic mode. In *predef3s* there are only two path options between nodes 0 and 1: the direct lower-delay path preferred by EIGRP that results in higher drop counts, and the higher-datarate path traversing node 2 preferred by OSPF that results in higher delay values. While the GNN policy learns to imitate EIGRP, the MLP learns to imitate both.

### C.3    Individual Random Graph Generators

Figures 13, 14 and 15 show the results of learning runs on random 10-node graphs (i.e. scenario *nx10*) obtained only on one of the three mentioned random graph models (BA, ER, WS). While the GNN policy performs somewhat similarly on all three graph generators, there is a higher ratio of seemingly "hard" instances among the WS graphs.

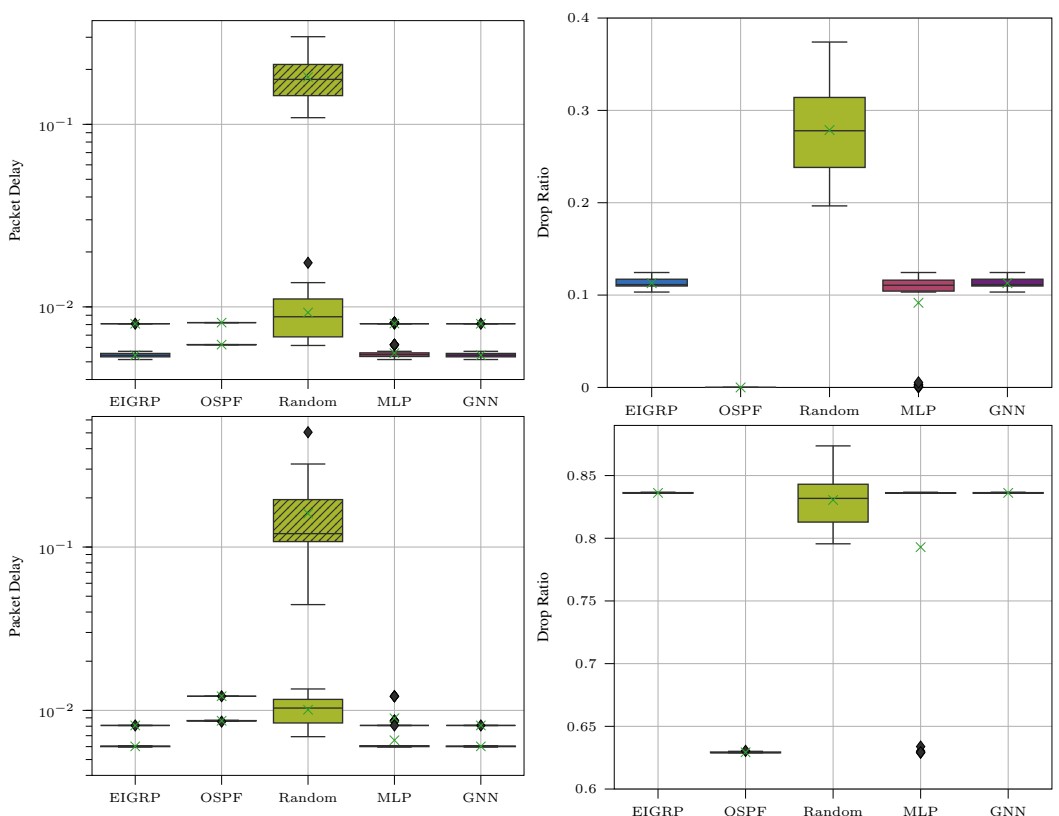

Figure 12: Average and maximum packet delay (left) and dropped packet ratio (right) per episode for experiments on *predef3s* in *flat* (top) and *peak* (bottom) traffic mode.

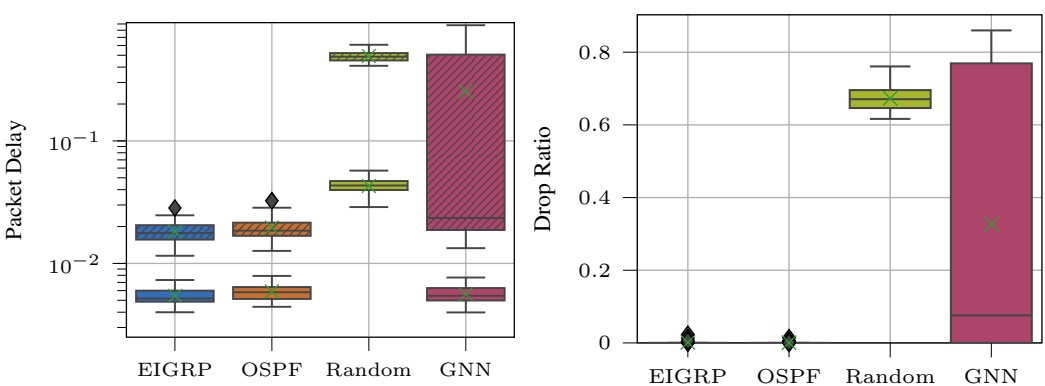

Figure 13: Average and maximum packet delay (left) and dropped packet ratio (right) per episode for experiments on 10-node BA graphs.

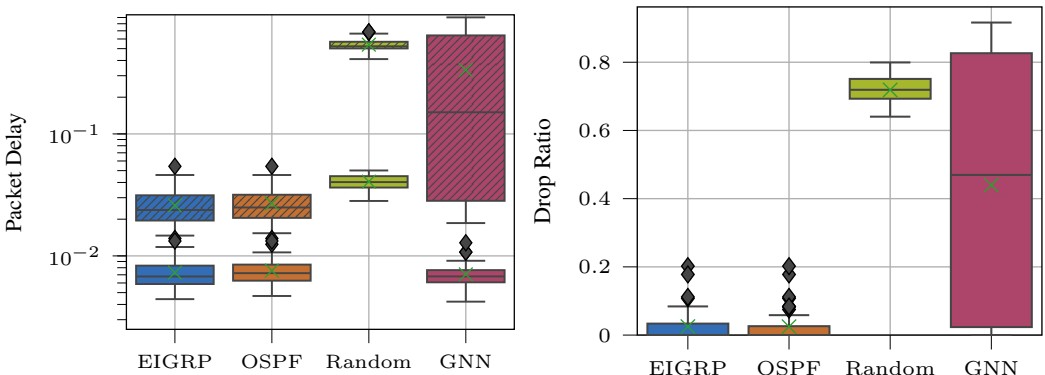

Figure 14: Average and maximum packet delay (left) and dropped packet ratio (right) per episode for experiments on 10-node ER graphs.

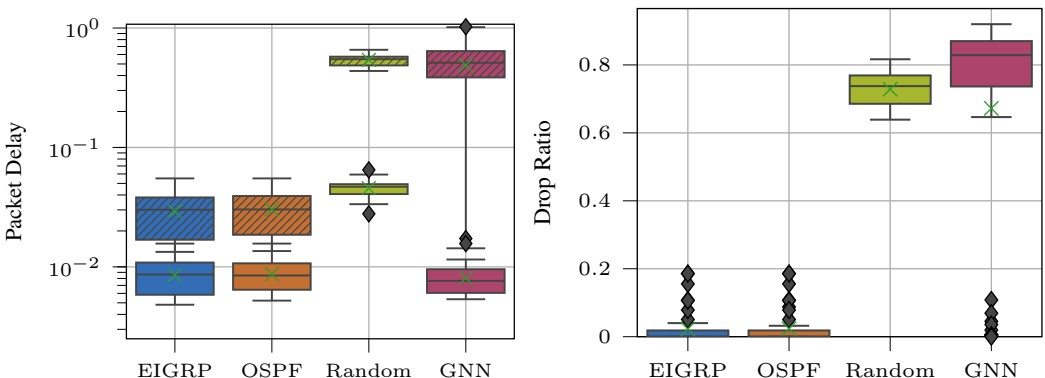

Figure 15: Average and maximum packet delay (left) and dropped packet ratio (right) per episode for experiments on 10-node WS graphs.

# D  ABLATION STUDIES

We report results for additional experiments that represent ablation studies. All experiments are run on the predefined 5-node network *predef5*. For default hyperparameter values, see B.

## D.1  ARCHITECTURAL ABLATIONS

In addition to the two policy variants introduced in section 5.1 we implemented additional policy variants $\phi_{\text{MLP+}}$, $\phi_{\text{GNN-}}$ and $\phi_{\text{Att}}$. For $\phi_{\text{MLP+}}$, we split the output of $\phi_{\text{MLP}}$ per (destination) node and feed each output part into a shared additional component consisting of two dimensionality-preserving linear layers and LeakyReLU activations after each. The GNN ablation $\phi_{\text{GNN-}}$ equals $\phi_{\text{GNN}}$ except that no auxiliary node features containing relative distances are provided. Finally, $\phi_{\text{Att}}$ implements an attention-like mechanism loosely inspired by Vaswani et al. (2017). For each combination of edge $e$ and destination node $j$, it applies an MLP to the concatenated features vectors of $e$, $j$ and the global features, followed by a readout layer to obtain $|V| \cdot |E|$ scalar score values. Furthermore, $\phi_{\text{GNN}_{\text{OSPF}}}$ is a GNN policy that uses OSPF instead of EIGRP for obtaining auxiliary node features. Finally, $\phi_{\text{MLP}_{\text{edge}}}$ and $\phi_{\text{GNN}_{\text{edge}}}$ utilize baseline variants without the final mean operation, yielding an individual value estimate for each edge.

Figures 16 and 17 show the results for these policy architecture ablations. $\phi_{\text{GNN}_{\text{OSPF}}}$ can perform similarly well than the default GNN policy, however its average performance is worse due to a higher number of episodes on which it is performing poorly. The architecture ablations $\phi_{\text{MLP+}}$ and $\phi_{\text{Att}}$ perform better than the random policy but far worse than the other NN policies. Interestingly, $\phi_{\text{GNN-}}$ performs competitively in terms of delay, but does to by learning to drop even more packets than the random policy.

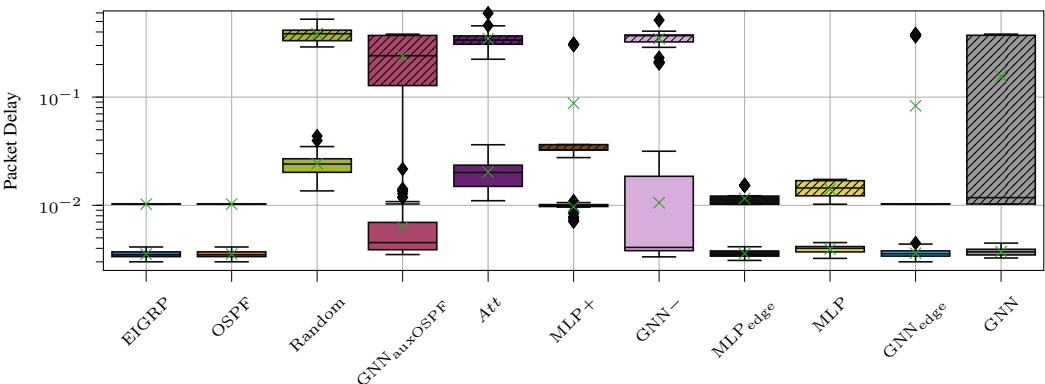

Figure 16: Average and maximum packet delay per episode for architecture ablations on *predef5*.

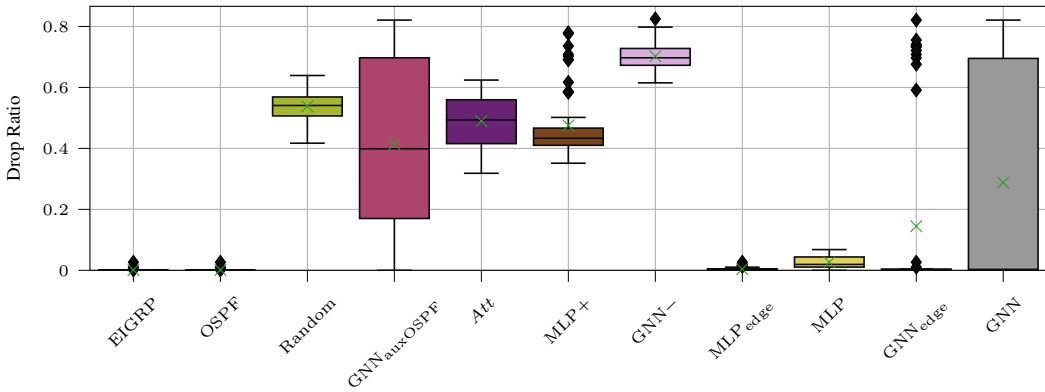

Figure 17: Drop ratio per episode for architecture ablations on *predef5*

## D.2 REWARD FUNCTIONS

In addition to the weighted delay $R^{\mathrm{wd}}$ and the drop ratio $R^{\mathrm{dr}}$, we implemented four other reward functions that are evaluated in this section:

$$R_t^{\mathrm{LU}}(s, a) = \max_e \mathtt{LU}(e)$$

is the maximum link utilization,

$$R_t^{\mathrm{ad}}(s, a) = \frac{1}{|P_t|} \left( \sum_{p \in P_t} d(p) \right)$$

is the unweighted average delay,

$$R_t^{\mathrm{md}}(s, a) = \max_{p \in P_t} d(p)$$

is the maximum delay,

$$R_t^{\mathrm{loop}}(s, a) = \frac{\mathtt{cycles}(a)}{|V|}$$

is the average routing loop count per node. These reward functions come with their own scaling coefficients $\rho_{\mathrm{LU}}$, $\rho_{\mathrm{ad}}$, $\rho_{\mathrm{md}}$ and $\rho_{\mathrm{loop}}$ for which the default values can be found in section B.4.

Figures 18 and 19 show the results for learning setups using $\phi_{\mathrm{MLP}}$ that each utilize only a single reward component, as well as a setup that uses $R^{\mathrm{comp}} = \rho_{\mathrm{wd}} R^{\mathrm{wd}} + \rho_{\mathrm{dr}} R^{\mathrm{dr}} + \rho_{\mathrm{loop}} R^{\mathrm{loop}}$. Besides the default setup $R = \rho_{\mathrm{wd}} R^{\mathrm{wd}} + \rho_{\mathrm{dr}} R^{\mathrm{dr}}$, only $R^{\mathrm{wd}}$, $R^{\mathrm{dr}}$ and $R^{\mathrm{comp}}$ show comparable performance, with $R^{\mathrm{dr}}$ improving on the drop ratio metric. However, we note that the default function $R$ yields the most consistent results on both metrics.

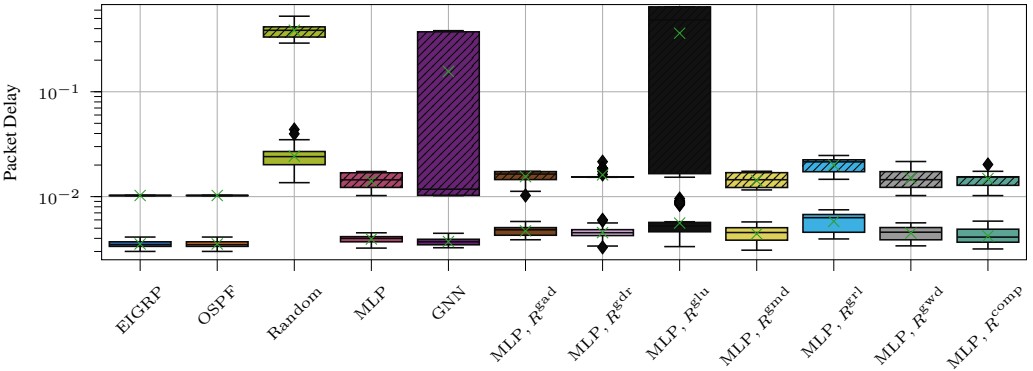

Figure 18: Average and maximum packet delay per episode for reward function ablations on *predef5*.

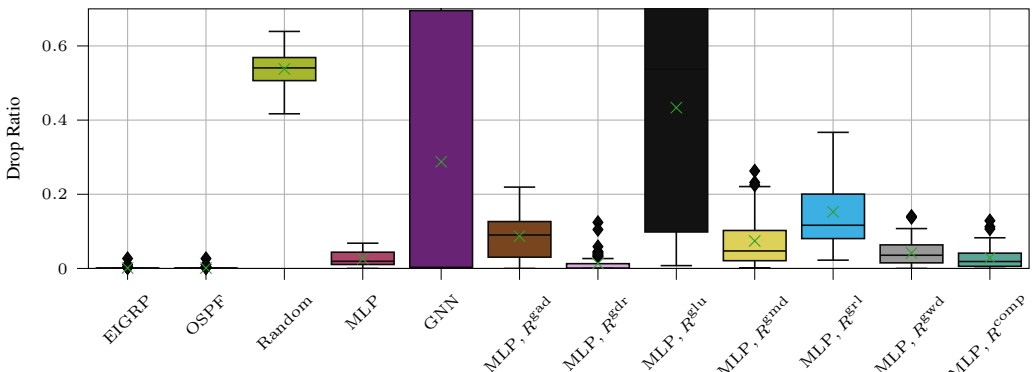

Figure 19: Drop ratio per episode for reward function ablations on *predef5*.

### D.3 EXPLORATION MECHANISMS

We experiment with a deterministic actor component that directly outputs $A'_t$ instead of the mean for a diagonal Gaussian (we denote this variant by $\phi_{\text{det}}$ while the original probabilistic actor is called $\phi_{\text{prob}}$. Also, we implement an alternative to the $\arg\max$-based assignment of $\psi$ by instead treating the values $\{A'_{ej} \mid e = (u,v), j \in V, v \in \mathcal{N}_i\}$ per routing node $i$ per destination $j$ as probabilities of a categorical distribution, and obtaining the gateway preferences $A_i$ from sampling these categorical distributions (we denote this variant by $\psi_{\text{cat}}$ as opposed to $\psi_{\text{argmax}}$. Figures 20 and 21 show the results for learning setups with alternative exploration mechanisms, indicating that the ablations do not improve the performance of our policies.

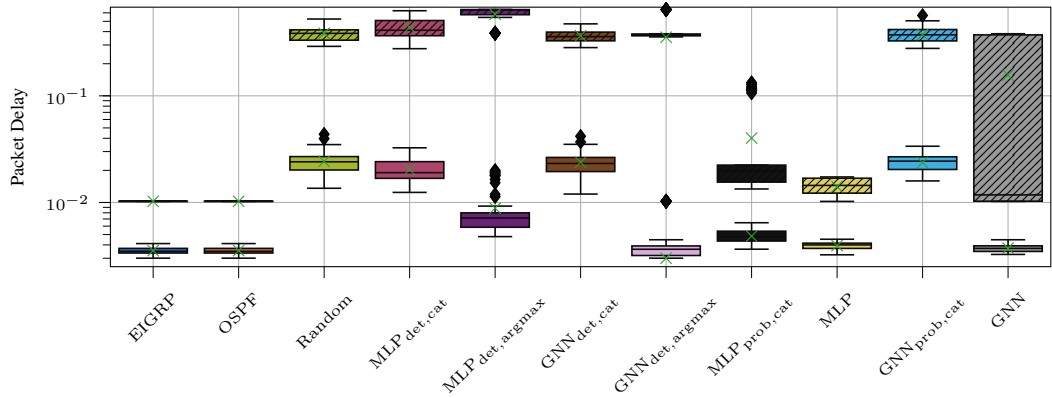

Figure 20: Average and maximum packet delay per episode for exploration mechanism ablations on *predef5*.

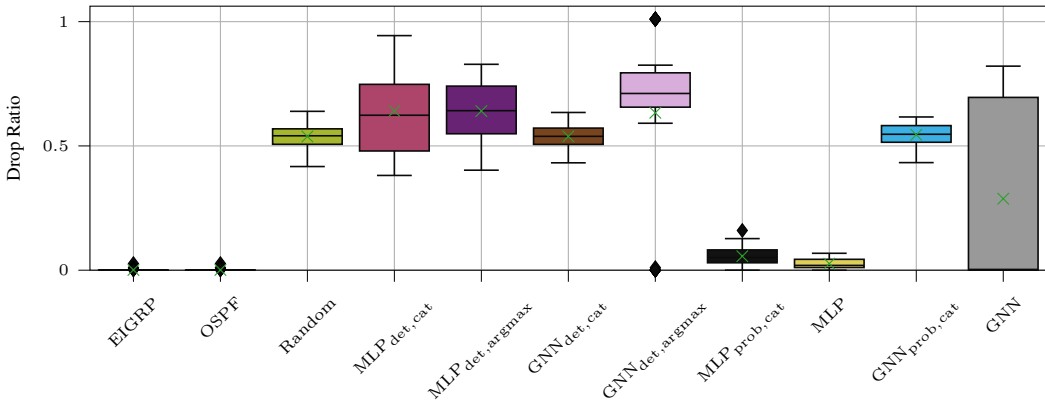

Figure 21: Drop ratio per episode for exploration mechanism ablations on *predef5*.

### D.4 PPO PARAMETERS

Figures 22 and 23 show the results of learning runs using $\phi_{\text{MLP}}$ with deviating PPO hyperparameters, namely $\gamma \in \{0.90, 0.99\}$ and $\alpha = 3e\text{-}4$. While a higher learning rate introduces training instability, variations in the discount factor lead to slightly higher worst-case performances.

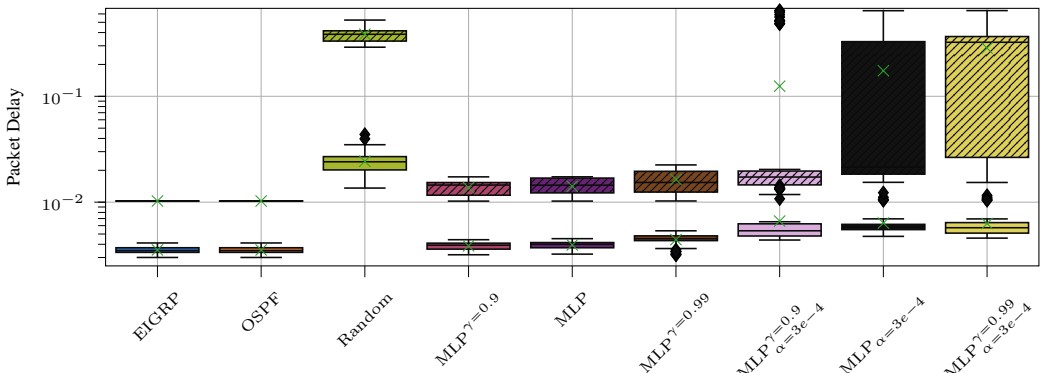

Figure 22: Average and maximum packet delay per episode for PPO ablations on *predef5*.

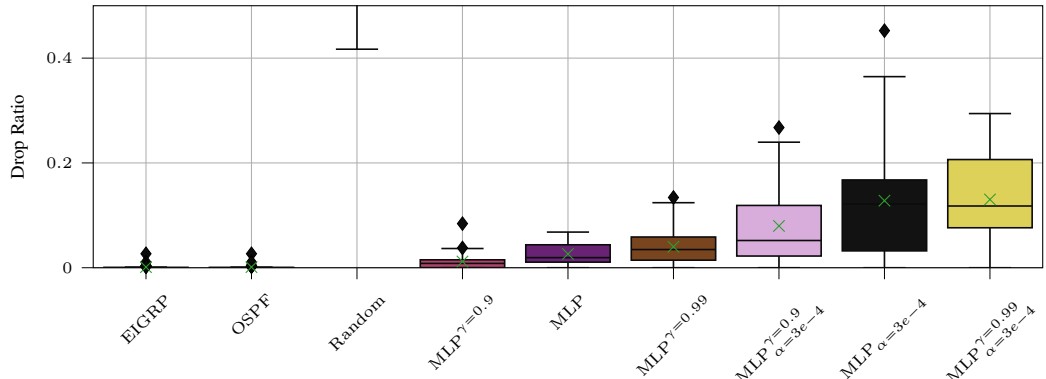

Figure 23: Drop ratio per episode for PPO ablations on *predef5*.

