# OpenReview forum: "Beyond Shortest-Paths: A Benchmark for Reinforcement Learning on Traffic Engineering"
_ICLR.cc/2024/Conference — Submitted to ICLR 2024_

### Official Review · Reviewer_XHPP · 2023-10-31

**Soundness:** 2 fair
**Presentation:** 2 fair
**Contribution:** 1 poor
**Rating:** 3
**Confidence:** 4

**Summary:**

This paper proposes an open-source framework named eleganTE, which aims to better solve traffic engineering (TE) with reinforcement learning (RL) methods in changing network states. The framework eleganTE fully uses the extension modules of ns-3 to provide realistic network environments. Based on this framework, the authors propose to use Markov Decision Process to formulate TE and use SwarMDP method to train RL agents for TE decisions. Evaluation results show that the proposed RL-based algorithm can outperform popular shortest-path algorithms.

**Strengths:**

1. In the proposed framework, eleganTE, authors use the GNN structure for RL-based TE framework, which can generalize to the previously unseen topologies, and it has the great potential to deal with some unexpected topology changes situations.

2. The authors use three extension modules in ns-3 simulator to monitor and capture the topology graph, generate realistic traffic, and simulate complex routing protocol situations, which seem to obtain more realistic network situations compared to other RL-based TE works using simulators.

**Weaknesses:**

1. Lack of novelty and insights. The authors use the RL-based method to solve TE and rely on discrete event-based network simulators (ns-3) to provide realistic network environment, which is widely explored in existing RL-based TE works, and it is not a new idea. Also, the authors list the requirements of TE that eleganTE solves in section 3, and they are also widely-known features in modern TE systems, which are not new insights. The three extension modules of ns-3 seem to be the new features in eleganTE compared to other RL-based TE works, but these features are only used to provide a realistic network environment, and they are irrelevant to the TE solution itself.

2. The motivation of the proposed methods is unclear. In section 4, the authors mention eleganTE uses MDP to formulate TE, but do not describe what this new formulation is and why it is superior to existing TE formulation ways. In addition, the time-slicing method that the authors proposed is also common in TE systems. Also, when the authors introduce SwarMDP to the framework, they do not mention the structure and advantages of SwarMDP, and the reason why it is superior to the existing methods.

3. Lack of comparison with the state-of-the-art RL-based TE works. The authors only compare eleganTE with the most naive TE methods, such as OSPF and EIGRP. As the authors have mentioned, there are a lot of recent RL-based TE works that have good performance, and authors need to compare them with these works to make the proposed framework more convincing.

4. Lack of exploration of common TE objectives. The TE objective explored in this work is the packet delay and packet drop ratio. There are more common TE objectives in modern TE systems, such as maximum link utilization (MLU) and throughput, and the authors need to add these evaluations.

**Questions:**

1. My first question is the novelty of the authors' work and what new contributions have been made to the RL-based TE. The authors claim that they propose a new open-source framework for RL-based TE, but the method is similar to most of the previous RL-based TE works, which use RL learning agents to make TE decisions and use network simulators (such as ns-3) to provide network environment. The three custom ns-3 modules seem different from common RL-based TE works, but in the evaluations, the authors do not show the superiority of these modules regarding solving RL-based TE.

2. In the abstract and the section of TE requirements (section 3), the authors aim to solve the challenges in networks with changing states, but this topic and the proposed challenges seem to be well solved by existing RL-based TE works. So the authors may need to gain new insights and propose relevant inspiring solutions.

3. I am also very confused about the formulation of the RL-based TE algorithm itself. First, the authors propose a new formulation for TE, named Markov Decision Process, without describing its detail and explaining why it is superior to existing TE formulations. Also, the advantages of SwarMDP over other RL algorithms are not explained in detail.

4. In the evaluations, I think the authors mainly need to compare other state-of-the-art RL algorithms or compare the realistic TE results with and without using the proposed framework, but they both lack in the authors' paper. Instead, the authors compare naive OSPF and state the proposed framework can outperform OSPF both in the abstract and conclusion. In addition, when it comes to the traditional TE method, MPLS methods seem to be superior to naive OSPF, the authors should also compare with it if they mainly want to compare traditional methods.

5. The last issue I want to mention is similar to what I have stated in the Weaknesses part, that is, minimizing MLU and maximizing throughput are more common TE objectives in modern systems. The authors may also need to evaluate these objectives.

---

> ### Author Response · Authors · 2023-11-19
> **Response #1.1 to Reviewer XHPP**
>
> [1/2]
>
> We thank you sincerely for your review, which exposes some important aspects we should work on as well areas where should improve the presentation of our findings. We address your concerns by topic (Weakness=W, Question=Q):
>
> **Novelty of the eleganTE framework (W1/Q1)**: While it is true that several related works have used ns-3 or other simulators/emulators to evaluate similar scenarios, we would like to emphasize that these are either not openly available, or restricted to a specific scenarios. A core part of the novelty of eleganTE is the combination of providing versatile and repeatable scenarios with an arbitrary number of graph topologies and traffic patterns, and making it fully available for fellow researchers as a new benchmark suite. The three custom ns-3 modules are necessary extensions to the simulator to faithfully implement the SwarMDP formalism proposed in Section 4 (now Section 3) and thus their roles in implementation are briefly presented in Section 5 (now Section 4), while we leave a more detailed explanation of eleganTE’s inner workings to Appendix A. As reviewer 3jBT has made similar remarks, we acknowledge that we did not present the limitations of related work and the unique position of eleganTE clearly enough, and have revised the structure and improved the presentation of the first three chapters to better motivate our framework.
>
> **Novelty of / Motivation for the SwarMDP formulation, State of related work (W2/Q2/Q3)**: We intend to introduce the requirements for RO suitable for TE, and then highlight how **only** our proposed formalism is able to perspectively fulfill all of them. We fully agree with your remarks and realize that we did not link existing RL-based TE work to the stated requirements well enough. We have revised the introduction, related work and requirements section (and integrated the requirements into the introduction) to better expose the conceptual weaknesses of existing work, and the proposed improvement of our formalism. Furthermore, we have improved the presentation of our formalism’s advantages in what is now Section 3.
>
> **Comparison in evaluation with related work (W3/Q4)**: Most of the mentioned related RL-based TE approaches do not warrant a comparison due to their limitations with respect to our requirements, but we do acknowledge that since we did not highlight the conceptual shortcomings of related work clearly enough (see the previous paragraph) it seemed as if the mentioned related work is directly comparable to our proposed approach. Furthermore, most related work lacks a publicly available reference implementation and/or detailed enough description of the algorithm and evaluation procedure to warrant an honest comparison. We will modify Section 2 to highlight this fact. However, despite the conceptual differences, we aim at including a comparison against [1] for reference in future work due to their remarkable results. Concerning non-RL approaches, we would like to point out that label-based approaches like MPLS or Segment Routing (SR) introduce additional layers, components and dependencies into the networking system stack that possibly counteract the idea of a general-purpose TE approach in the face of heterogeneous network types and hardware setups (c.f. requirement “Compatibility”). On the other hand, our odd-routing module with RL-based routing serves as a drop-in replacement for Routing Protocols like OSPF or EIGRP. Nevertheless, we agree that comparison to non-RL TE approaches should not be neglected, and will provide a reference comparison to modern label-based TE approaches [2, 3] in future work.

---

> ### Author Response · Authors · 2023-11-19
> **Response #1.2 to reviewer XHPP**
>
> [2/2]
>
> **Evaluation TE objectives (W4/Q5)**: We agree on MLU and throughput being common TE objectives and would like to explain our motivation for using the metrics you find in our evaluation: Most of our paper’s experiments concern UDP traffic in the _peak_ traffic mode, which has been chosen to mimic stress scenarios in networks where certain periods of congestion are unavoidable when using OSPF or EIGRP (i.e. the maximum LU over an episode is always 1.0). We have made this fact more clear in Section 5 and Appendix A.4. Since UDP in its base form simply sends out packets with the full desired sending rate and all parallel evaluations of OSPF, EIGRP, GNN etc. use the same traffic matrices, the amount of sent packets in each parallel episode is effectively the same (of course, it can change between episodes). Therefore the main indicators of an improved routing capability in these scenarios are fewer packet drops which in the case of “fixed” sent packet counts is fully correlated with throughput, and lower packet delay. Concerning the evaluations on TCP traffic, TCP controls the datarate of the sender applications to minimize congestion, and so in addition to the amount of dropped packets (because TCP is not able to prevent them all) throughput indeed is a relevant performance metric. In Figure 7 we chose to display the amount of sent packets over the entire episode as an episode-wide average of the throughput, next to the amount of dropped packets as an indicator of data actually reaching its goal and not just being distributed into the network nicely.
>
> References:
>
> [1] Bernárdez, Guillermo, et al. "MAGNNETO: A Graph Neural Network-Based Multi-Agent System for Traffic Engineering." IEEE Transactions on Cognitive Communications and Networking 9.2 (2023): 494-506.
>
> [2] Gay, Steven, Renaud Hartert, and Stefano Vissicchio. "Expect the unexpected: Sub-second optimization for segment routing." IEEE INFOCOM 2017-IEEE Conference on Computer Communications. IEEE, 2017.
>
> [3] Jadin, Mathieu, et al. "Cg4sr: Near optimal traffic engineering for segment routing with column generation." IEEE INFOCOM 2019-IEEE Conference on Computer Communications. IEEE, 2019.

---

### Official Review · Reviewer_3jBT · 2023-11-01

**Soundness:** 3 good
**Presentation:** 2 fair
**Contribution:** 2 fair
**Rating:** 3
**Confidence:** 4

**Summary:**

The objective of this paper is to utilize a swarm reinforcement learning approach towards route optimization in networks.  The reward function is based on the delay incurred by packets, with a drop being considered as  maximum possible delay.  The paper presents empirical results over a simulator that shows good performance against existing approaches.

**Strengths:**

The  paper considers an important networking problem and tailors a ML approach towards its solution.

**Weaknesses:**

The multi-agent RL portion is poorly described.  It appears that the author simply utilize the multi-agent toolboxes available, with an appropriate state, action, reward functions.  As such, I am  not clear as to the level of ML related contributions in this paper.  It looks like a use case of an existing approach, with small modifications to the approach and the ns3 simulator appropriate the situation.

**Questions:**

- It appears that the traffic, although dynamic, is essentially fixed for periods of time where routing optimization is performed.  Is this correct?
- It is not clear what the full set of observations is.  Does it include the traffic demand on each node?  If so, how is it measured?

---

> ### Author Response · Authors · 2023-11-19
> **Response #1 to Reviewer 3jBT**
>
> We thank you very much for your review and for exposing the need for improvement in presentation clarity.
>
> > The multi-agent RL portion is poorly described. It appears that the author simply utilize the multi-agent toolboxes available, with an appropriate state, action, stewards functions. As such, I am not clear as to the level of ML related contributions in this paper. It looks like a use case of an existing approach, with small modifications to the approach and the ns3 simulator appropriate the situation.
>
> As part of our contribution, we intend to expose the conceptual weaknesses of related work by first stating the requirements that we deem critical for RO that can work for autonomous TE, and then provide an MDP formulation that in contrast to previous work lends to RO mechanisms that fulfill all stated requirements. We agree with your statement in that our work does not clearly motivate the MDP formulation and its uniqueness, and have considerably improved its presentation to motivate its need and novel features more clearly. While it is true that our TE SwarMDP is similar to the formulations of the cited papers, our action space formulation permits agents with varying neighbor counts without losing the homogeneous nature of their respective action spaces. We have also adjusted the section (which is now Section 3) to present this improvement more clearly. Furthermore, a core contribution of this paper is the disclosure of eleganTE, a versatile framework that enables repeatable experiments within this formalism. Given the shortcomings and non-comparability of related work, we believe eleganTE complements our formalism in a way that is of great value for the ML x Computer Networking community.
>
> > Q1. It appears that the traffic, although dynamic, is essentially fixed for periods of time where routing optimization is performed. Is this correct?
>
> For our answer we assume that by “periods of time where routing optimization is performed” you refer to the “execution” of the timestep in the simulator (please let us know if we have misunderstood you here). The traffic behavior is “fixed” in the way that, at the start of each simulation period (i.e. environment step), the implementing _demand-driven-applications_ adopt a constant sending bitrate so that, over the course of \tau_{\text{sim}}, an amount of data is injected into the network that matches the corresponding entry in the traffic matrix. We will improve the wording in the framework Section (now Section 4) to clarify that the bitrate is constant throughout a timestep. As the concept of traffic matrices themselves is a discretization of the continuous time-series nature of traffic dynamics, it is an interesting question how to achieve a fully variable sending rate without moving away from using traffic matrices altogether.
>
> > Q2. It is not clear what the full set of observations is. Does it include the traffic demand on each node? If so, how is it measured?
>
> We observe network device load values and link utilization values locally for each edge. These values range from 0 (no load/usage) to 1 (full network device buffer/fully utilized link). Also, we observe global maximum link utilization and average bandwidth utlilization, which we by default store in each node. As traffic patterns in modern enterprise networks can be highly volatile and inherently unpredictable [1], we choose to not include traffic statistics directly in our observation function but rather via the observed utilization values. This means that we treat the traffic dynamics as part of the unknown transition function of our SwarMDP for which we would like to find an approximately optimal solution using RL. The observation procedure is explained in Appendix B.2, and we will add a brief note in teh formalism Section to improve the section’s clarity of presentation.
>
> References:
>
> [1] Gay, Steven, Renaud Hartert, and Stefano Vissicchio. "Expect the unexpected: Sub-second optimization for segment routing." IEEE INFOCOM 2017-IEEE Conference on Computer Communications. IEEE, 2017.

---

### Official Review · Reviewer_dSxa · 2023-11-02

**Soundness:** 3 good
**Presentation:** 3 good
**Contribution:** 3 good
**Rating:** 6
**Confidence:** 3

**Summary:**

The paper targets the problem of selecting efficient routes for data packets, where optimal routes depend on the current network state which are very dynamic.

The paper contributes by framing the distributed Traffic Engineering as a Swarm Markov Decision Process, and contributes a training and eval framework - eleganTE supported by a reliable network simulation engine.Through simulation, the paper shows the effectiveness of the framework, and how it outperforms the popular shortest-path routing algorithms.

**Strengths:**

- Tackles a hard problem at the intersection of networking and artificial intelligence
- The proposed framework eleganTE facilitates repeatable experiments on network scenarios with a large variety in topology and traffic patterns
- The presented policies match or outperform popular shortest path RPs
- Contributes an approach that is a step towards automating computer networks, can improve OE, save costs, and can be expanded to transport networks or power grids.
- Contributes by defining the requirements that RO techniques must fulfil to be effective for TE in practice

**Weaknesses:**

- The framework does not support the design of scenarios with changing topologies and corresponding policies as of now.
- Training stability is an issue.
- Does not support decentralized training and execution paradigm which is necessary for a truly distributed TE
- Does not evaluate the policies on real world networks, beyond simulation.

**Questions:**

N/A

---

> ### Author Response · Authors · 2023-11-19
> **Response #1 to Reviewer dSxa**
>
> We thank you for your review and the acknowledgement of the strengths of our work. We agree on the listed weaknesses and, as indicated in the paper, will work on resolving them in the future. Concerning the training stability issue, we would like to emphasize the role of implementation details when using the widely popular PPO algorithm [1]. Given the focus of our work, it is conceivable that despite the executed hyperparameter ablation studies available in Appendix D, it is conceivable that the instability can be reduced by optimizing the learning algorithm implementation, and we have stated this fact more clearly in the conclusion.
>
> References:
>
> [1] Engstrom, Logan, et al. "Implementation matters in deep policy gradients: A case study on ppo and trpo." arXiv preprint arXiv:2005.12729 (2020).

---

### Official Review · Reviewer_HzR2 · 2023-11-03

**Soundness:** 2 fair
**Presentation:** 3 good
**Contribution:** 2 fair
**Rating:** 3
**Confidence:** 5

**Summary:**

The paper introduces eleganTE, a framework for the efficient training and evaluation of routing algorithms that are learned via reinforcement learning RL). The framework relies on the ns-3 discrete-event network simulator, which provides a faithful simulation as opposed to other frameworks that assess routing performance based on the abstract network graph. Furthermore, it provide a rich generation process for various network topologies and traffic patterns. On the theory side, the authors cast routing optimization as a Swarm RL problem, which in principle allows them to train policies on the node level that can generalize to unseen network topologies. The authors assess their framework on various diverse scenarios, and claim that  in some cases the learned routing strategy via RL can outperform the popular shortest-path routing algorithms such as OSPF and EIGRP.

**Strengths:**

1. The authors correctly identify the necessity of faithful simulations based on state-of-the-art network simulators, since these can better take into account interference effects, protocol interplay, delays etc. Their framework is built upon such a realistic simulator, which can better and more accurately assess the relative performance of different routing strategies. The proposed framework can be of high value to researchers working in the intersection of ML and networking.
2. The proposed framework can generate a variety of random network topologies (BA, ER model and WS). Furthermore, it uses the gravity model for the traffic matrix, together with small random perturbations to cover a variety of traffic dynamics.
3. The authors provide in Appendix D various ablation studies, covering the policy architectural choice, the reward function choice, and even the actor component design.

**Weaknesses:**

I have various concerns about the current paper.
1. The extensive experiments conducted by the authors seem to universally suggest that the relatively simple shortest-path-based protocols OSPF and EIGRP generally outperform MLP and GNN, and usually by large margins. The authors claim that in some cases the RL-based methods can outperform the standard protocols; as an example, they mention the predef4s topology in Figure 4. However, even though the packet delay is indeed slightly lower for MLP in Figure 4, the drop ratio is visibly higher, effectively canceling out the packet delay benefit. Results do not look promising at all for random topologies, see, e.g., Figure 6, where the max delay as well as the drop ratio for GNN can be particularly high.
2. To me, the experimental evaluation fails to show how RL-based routing optimization makes sense in the first place. There is not a single setting, where MLP or GNN clearly beat the standard protocols in terms of both metrics. This inevitably casts doubt on the necessity of the framework. Is it for example possible that in practice shortest-path algorithms such as OSFP can easily adjust to the changing traffic by simply updating the shortest path based on the observed congestion, delays etc.? Demands contain a strong stochastic component, so is it possible that simply reacting to the changing network condition via the standard protocols is as effective (if not more effective) as complex and harder to deploy ML-based techniques? I do not think that the current work provides any encouraging answer (in favor of ML-based approaches) in this direction.
3. I am not sure that formulating routing optimization as a swarm RL problem is well justified. SwarmMDP typically assumes a swarm of homogenous collaborating agents. But is this assumption true in flow networks? One major concern is that for the relatively small network topologies (e.g., with 3-10 nodes) that the authors have experimented with, nodes may be better modeled as heterogeneous (unless the network topology is fully symmetric, as in the complete graph). For instance, a high-degree node is expected to behave very differently compared to a peripheral node. Using GNNs makes perfect sense, but the swarm assumption seems harder to justify in tis context.
4. The problem may be better modeled as a multi-objective RL problem, if we are interested in multiple metrics, e.g., packet delay and drop ratio. The authors employ a scalarized reward, but this seems quite ad-hoc. In Appendix B.4, for example, the authors mention that they set the reward scaling factors to specific constants, but it is not clear why that choice makes sense. Furthermore, in the ablation study in Appendix D.2, results are reported on a single network topology (predef5). This is not enough evidence to draw the conclusion that the reward function in the main text is generally better than the other reward functions. There may in principle be topologies where some of the other reward functions can achieve superior performance (with a good hyperparameter search).
5. It seems that no real network topologies are used. Furthermore, the random topologies have up to 50 nodes. I am not clear whether this is in accordance with the desideratum of scalability, or whether the framework should be assessed on even larger topologies for this purpose.

**Questions:**

1. Are there settings where RL-based techniques outperformed the standard protocols across both reported metrics by clear margins? If not, what does that suggest for the proposed framework but also for the necessity of RL-based routing optimization for communication networks?
2. Is SwarmMDP a meaningful approach, especially in the small network regime where nodes behave very differently (asymmetrically), depending on their position in the graph?
3. Would multi-objective RL make more sense? Can we be confident that the used hyperparameters and that the scalarized reward function in the main text are indeed the right choice for a wide range of networks? What about hyperparameter sensitivity?
4. Are the pre-defined topologies from real networks? If yes, maybe it would make sense to assess the framework on bigger real networks?
5. GNN seems to always suffer from bad performance. I am wondering if this contradicts the claim that the GNN has the advantage over the MLP that it can generalize to unseen topologies. In principle this may be true, but the results do not look encouraging.

---

> ### Author Response · Authors · 2023-11-19
> **Response #1.1 to Reviewer HzR2**
>
> [1/2]
>
> Thank you very much for your very detailed review and the various concerns you have rightly pointed out. You have mentioned many very valuable points that we would like to address by topic (Weakness=W, Question=Q):
>
> **Overall Performance vs. Standard RPs, Motivation for RL-based TE (W2/Q1)**: We fully agree that the performance of our own trained policies is not superior to the presented RP baselines. However, this does not mean that Routing Protocols like OSPF adjust to changing traffic. They use fixed equations to calculate shortest paths per source-destination pair, using configuration parameters like link datarate. Traffic Engineering, which includes adjusting “shortest paths” to respect traffic conditions, is achieved either by manually adjusting the link weights for such routing protocols in the face of traffic changes (which is obviously highly inefficient at scale), or by technologies such as Segment Routing (SR), MPLS or ECMP. These techniques however, as indicated in the RO requirements that now form part of the introduction (c.f. general response), introduce additional complexity into the overall network structure, and are not guaranteed to react adequately in previously unencountered situations since they operate on pre-defined heuristics, too. Here, a routing policy that is learned e.g. via RL, can cover an unprecedented breadth of situations, a promise that has been partially validated by related work. This paper’s main goal is to highlight the shortcomings of related work, and then introducing a formalism and an implementing framework which facilitate the training of policies that can be used for real-world autonomous TE without the shortcomings of related work. We believe that this alone is already a notable contribution, combined with the fact that related work has shown the superiority of RL-based approaches for select TE use cases. In fact, we have provided the admittedly sub-optimal policies mainly as a demonstration of the framework’s features and of the difficulty of the implemented decision problem, and we are confident that future work building on our framework (our own as well as others that might use it) will show substantially improved performance and clearly outperform existing baselines. However, we fully acknowledge that the existing structure of the paper does not motivate the need for eleganTE well enough, and thus we have revised the first few sections to better highlight its novelty and value (c.f. general response).
>
> **SwarMDP vs. Heterogeneous Agents (W3/Q2)**: In small networks we might get away with modeling our routing agents heterogeneously, however when scaling the network size the complexity of problem formulations with heterogeneous agents will grow uncontrollably due to its combinatorial nature. We believe that the optimal behavior of a node can be derived from its input features (i.e. local connectivity and utilization statistics) alone. As a consequence, routing nodes are well modeled as homogeneous agents that differ only in the varying amounts of neighbors, which we respect with our action space formulation.
>
> **Figure 4/Outperforming Scenario (W1)**: Concerning the results on predef4s, we realize that the employed method of visualization is not accurate enough to showcase the performance of our NN methods, especially with respect to the claim of “outperforming” OSPF and EIGRP. The main goal of the presented policies is to showcase the versatility of eleganTE and to expose the limitations of conventional Routing Protocols in certain situations, and we hope to add routing visualizations that further support the claim made with respect to predef4s by the end of this rebuttal period.
>
> **Topologies and Scalability (W5/Q4)**: The pre-defined topologies are synthetic, and we would like to extend our topology generation procedure to include bigger real networks e.g. from the TopologyZoo [1] in future work. However we are confident that our randomly generated topologies already cover a wide variety of conceivable networking scenarios due to the multiple supported random graph models. Concerning the scalability requirement, our framework eleganTE is only practically bounded in network size due to a quadratic scaling of sender application count and size of the shared memory used for communication. As reviewer Jksb has also requested results on larger networks, we hope to extend our evaluations on random graphs by evaluations on randomly generated 150- and 500-node networks to support this statement.

---

> ### Author Response · Authors · 2023-11-19
> **Response #1.2 to Reviewer HzR2**
>
> [2/2]
>
> **General GNN Performance (Q5)**: We are aware of the high variance of GNN performance in our experiments, but would like to point out that the GNN was able to achieve comparable performance at least occasionally in the experiments. Combined with the results of related work, it is clear to us that GNNs can be a viable model architecture choice, especially since in contrast to MLPs they are able to handle graph-shaped inputs while not relying on a particular ordering of the input features. We do not think that the shown performance contradicts the claim, however we acknowledge that we have to considerably improve the reliability of our trained GNN policies, e.g. via implementation improvements in future work.
>
> **Multi-Objective RL (W4/Q5)**: Network performance is assessed in several different metrics that sometimes cannot be optimized without trade-offs. As such, we fully agree that using Multi-Objective RL (MORL) is a very interesting Idea and might be a better fit for our TE problem in the long run. However, Multi-Agent Multi-Objective RL is highly complex, still largely uncharted research area [2], not least because it is unclear how to weight the different common TE objectives since they themselves are proxies for performance metrics that network users actually care about [3]. Given the novelty of the framework as well as the difficulty of the considered problem even with single-objective RL, for this work we have decided to run with the approximate solution of scalarized rewards and fixed scaling factors. These scaling factors have been selected so that the expected average values for the individual scaled reward components lie in the same order of magnitude (and thus each corresponding objective is valued roughly equally), except for $R_\text{loop}$ which can range considerably higher the more loops have been introduced to the routing. We have added a short note in Appendix D.4 for improved clarity. Concerning the lack of evidence in the reward function ablations, we hope to add reward function ablation experiments on random graphs in future work.
>
> References:
>
> [1] Knight, Simon, et al. "The internet topology zoo." IEEE Journal on Selected Areas in Communications 29.9 (2011): 1765-1775.
>
> [2] Hayes, Conor F., et al. "A practical guide to multi-objective reinforcement learning and planning." Autonomous Agents and Multi-Agent Systems 36.1 (2022): 26.
>
> [3] Dukkipati, Nandita, and Nick McKeown. "Why flow-completion time is the right metric for congestion control." ACM SIGCOMM Computer Communication Review 36.1 (2006): 59-62.

---

### Official Review · Reviewer_Jksb · 2023-11-10

**Soundness:** 2 fair
**Presentation:** 2 fair
**Contribution:** 2 fair
**Rating:** 3
**Confidence:** 4

**Summary:**

The paper proposes a new benchmark for evaluating reinforcement learning (RL) methods for traffic engineering in computer networks. The key ideas and contributions are:
1.	Framing distributed traffic engineering as a Swarm Markov Decision Process (SwarMDP) where nodes collaborate to optimize network performance. This allows RL agents to generalize to new topologies.
2.	Introducing eleganTE, a framework for training and evaluating RL agents for traffic engineering using the ns-3 network simulator. The framework includes configurable topology and traffic generation, and interfaces for training RL agents.
3.	Evaluating common heuristics like OSPF and EIGRP routing as baselines, and comparing them to learned policies including MLP and graph neural network (GNN) architectures.

**Strengths:**

1.	The SwarMDP formulation described in section 4 provides a way to train policies that can generalize across network topologies, an important capability highlighted in Section 3.
2.	EleganTE in Section 5 enables training agents using a configurable network simulator, facilitating repeatable experiments.

**Weaknesses:**

1.	The SwarMDP formulation in Section 4 lacks details on handling heterogeneous action spaces and scaling to large network sizes.
2.	The eleganTE framework description in Section 5 needs specifics on the implementation, particularly how network state information is extracted into the monitoring graph representation.
3.	Experiments only evaluate small networks of up to 50 nodes in Section 7. Testing generalization performance to 500+ node networks is needed.
4.	The GNN agent in Section 7.2 shows high variance in some experiments, indicating instability issues.

**Questions:**

1.	Is the SwarMDP formulation described in section 4 able to handle changes in network topology within an episode?
2.	Provide more specifics of neural network architectures used in section 6.1
3.	Explore complex benchmark traffic patterns and dynamics in section 5.1
4.	The authors should explore algorithm ablations like reward function, network architecture, etc.

---

> ### Author Response · Authors · 2023-11-19
> **Response #1 to Reviewer Jksb**
>
> Thank you very much for your review and your raised concerns. We address them individually:
>
> > W1. The SwarMDP formulation in Section 4 lacks details on handling heterogeneous action spaces and scaling to large network sizes.
>
> We view the action spaces of the routing nodes as homogeneous, in the way that they all consist of unique gateway preferences per possible packet destination. This means that despite varying neighbor counts and different locations, all routing nodes fulfill the same task. We handle the varying neighbor counts with our proposed action space formulation, which therefore permits us to view this problem as one of homogeneous agents. Concerning the scalability to large networks, the proposed formalism is not constrained by network size. We have added a short clarification in the formalism Section (now Section 3), stating that it thus supports large networks.
>
> > W2 The eleganTE framework description in Section 5 needs specifics on the implementation, particularly how network state information is extracted into the monitoring graph representation.
>
> Due to page constraints, In the framework Section (now Section 4) we cover essential information about eleganTE’s implementation and refer to Appendix A for more extensive implementation details, including the monitoring procedure described in Appendix A.3. We will make this reference more clear in the paper.
>
> > W3 Experiments only evaluate small networks of up to 50 nodes in Section 7. Testing generalization performance to 500+ node networks is needed.
>
> eleganTE implements the SwarMDP described in the Formalism Section (now Section 3) and is only practically bounded in network size due to a quadratic scaling of sender application count and size of the shared memory used for communication. As reviewer HzR2 has also requested results on larger networks, we hope to extend our evaluations on random graphs by evaluations on randomly generated 150- and 500-node networks to support this statement.
>
> > W4. The GNN agent in Section 7.2 shows high variance in some experiments, indicating instability issues.
>
> As you have correctly pointed out the stability issues of the GNN in particular, we would like to emphasize the role of implementation details when using the widely popular PPO algorithm [1]. Since with our policies we primarily wanted to expose the challenging nature of this problem, it is conceivable that the instability can be reduced by optimizing the learning algorithm implementation, despite the already executed hyperparameter ablation studies available in Appendix D.
>
> > Q1. Is the SwarMDP formulation described in section 4 able to handle changes in network topology within an episode?
>
> While we will explore scenarios with changing network topology in future work, we think that the present SwarMDP formulation will be able to handle such changes with minimal modifications, i.e. substituting $V$ and $E$ with time-dependent counterparts $V_t$ and $E_t$. We have adjusted the notation to include this modification, with a note in the experiments Section (now Section 5) that currently, the topology stays constant.
>
> > Q2. Provide more specifics of neural network architectures used in section 6.1
>
> We agree in that Section 6.1 (now Section 5.1) does not cover the full details of the NN architectures. Due to page constraints, it only provides a brief overview of the used NN architectures, and we will make our reference to Appendix B.5, where we provide more architecture details, more clear.
>
> > Q3. Explore complex benchmark traffic patterns and dynamics in section 5.1
>
> We assume that by complexity you mean the variety of generated traffic patterns. We use the Gravity model [2] to create Traffic Matrices that mimic spiking traffic, and increase the variety (and thus complexity) of the generated traffic dynamics by introducing small random perturbations. Please let us know if this is not what you meant.
>
> > Q4. The authors should explore algorithm ablations like reward function, network architecture, etc.
>
> We have explored these ablations and provide their results in Appendix D.
>
> References:
>
> [1] Engstrom, Logan, et al. "Implementation matters in deep policy gradients: A case study on ppo and trpo." arXiv preprint arXiv:2005.12729 (2020).
>
> [2] Roughan, Matthew. "Simplifying the synthesis of Internet traffic matrices." ACM SIGCOMM Computer Communication Review 35.5 (2005): 93-96.

---

### Author Response · Authors · 2023-11-19
**General response to Paper Reviewers #1**

Dear reviewers,

we thank you for your very helpful responses to our work. In our view, your remarks and suggestions have already helped to improve the quality of our submission considerably. We have uploaded a temporary version of our paper in which we have included many of the reviewers' remarks in color while keeping the old text. For the final version at the end of the rebuttal period, we will of course remove the color and old text passages, returning to the main text length constraint of 9 pages. This general response briefly summarizes the changes visible in the temporary version, sorted by text color in the rebuttal:

- **Red text**: One of the main weak points exposed by reviewers XHPP and 3jBT is in our views the unclear focus of the paper. This includes the lackluster motivation for the presented formalism and eleganTE, the implementing framework: In the initial version we did not present clearly enough the conceptual and tooling-related gap that related work has left open, and therefore we have improved the wording and structure of the first half of the main text in particular to highlight how our SwarMDP formulation and eleganTE close the gap. Concretely, we have integrated the stated requirements for routing optimization into the introduction, and re-written parts of the related work section to better highlight the limitations of existing work with respect to the stated requirements. Then, the formalism Section which is now Section 3 introduces our SwarMDP formulation, and we have improved its wording to show more clearly how our formulation indeed caters to the requirements and, together with the implementing framework eleganTE, fills the gap left open by previous work. In fact, we view this as our paper's primary goal and a notable contribution on its own, while the presented policies' main purpose is to showcase the usefulness of eleganTE and to expose the challenging nature of the TE problem. As part of the overall set of contributions of this work, we acknowledge that the depicted results in fact do not clearly show that our policies outperform OSPF and EIGRP as stated (reviewer HzR2). We have thus adjusted our wording to highlight what they indeed do: to expose the weaknesses of the heuristic routing protocols in certain situations.

Apart from this bigger change in the paper, we have made several smaller adjustments:

- **Blue text**: further clarifications concerning remarks made by reviewer XHPP (maxLU being 1.0 in the _peak_ traffic mode),
- **Magenta text**: further clarifications concerning remarks made by reviewer 3jBT (formalism clarifications),
- **Orange text**: further clarifications concerning remarks made by reviewer HzR2 (reward scalarization),
- **Teal text**:  further clarifications concerning remarks made by reviewer Jksb (formalism clarifications),
- **Olive text**: minor improvements (wording, spelling, sentence structure), as well as moving the contents of former section 8.1 ("Broader Impact") into the optional ethics statement.

We thank the reviewers again for their comments and are eager to respond if questions remain.

---

### Author Response · Authors · 2023-11-22
**General response to Paper Reviewers #2**

Dear Reviewers,

we have uploaded an updated version of the paper (without the text color) that again adheres to the venue's 9 page main text limit. We are happy to respond if any questions remain.

---

### Meta-Review · Area_Chair_nNTH · 2023-12-07

**Metareview:**

The main strength/contribution of the paper is to introduce an evaluation framework for learning-based routing algorithms in computer networks. The main concerns/weaknesses about the paper are:
* There is no clear demonstration that RL-based routing outperforms simpler shortest-path baselines in the reported experiments
* Several aspects of the presentation were unclear (formulation, details of experimental setup, ...) in the initial submission. Even if these were improved in a revised version during the rebuttal the paper still needs substantial additional.
* The networks used in experiments are very small, and it isn't clear whether the proposed framework will scale to larger networks, which would be necessary for obtaining meaningful/useful conclusions

Because of these significant weaknesses the conclusion after discussing with the reviewers was to reject the paper.

**Justification For Why Not Higher Score:**

Because of the weaknesses listed above.

**Justification For Why Not Lower Score:**

N/A

---

### Decision · Program_Chairs · 2024-01-16

Reject